# BBCaL: Black-box Backdoor Detection under the Causality Lens

**Mengxuan Hu**[*]                                               *qtq7su@virginia.edu*
*University of Virginia*

**Zihan Guan**[*]                                                *bxv6gs@virginia.edu*
*University of Virginia*

**Junfeng Guo**                                                  *gjf2023@umd.edu*
*University of Maryland*

**Zhongliang Zhou**                                    *zhongliang.zhou@merck.com*
*Merck*

**Jielu Zhang**                                              *jielu.zhang@uga.edu*
*University of Georgia*

**Sheng Li**                                                     *vga8uf@virginia.edu*
*University of Virginia*

[*] *Contributed equally*

**Reviewed on OpenReview:** *https://openreview.net/forum?id=HZi9PfLwMn*

## Abstract

Deep Neural Networks (DNNs) are known to be vulnerable to backdoor attacks, where attackers can inject hidden backdoors during the training stage. This poses a serious threat to the Model-as-a-Service setting, where downstream users directly utilize third-party models (e.g., HuggingFace Hub, ChatGPT). To this end, we study *inference-stage black-box backdoor detection* problem in the paper, where defenders aim to build a firewall to filter out the backdoor inputs in the inference stage, with only input samples and prediction labels available. Existing investigations on this problem either rely on strong assumptions on types of triggers and attacks or suffer from poor efficiency. To build a more generalized and efficient method, we first provide a novel causality-based lens to analyze heterogeneous prediction behaviors for clean and backdoored samples in the inference stage, considering both sample-specific and sample-agnostic backdoor attacks. Motivated by the causal analysis and *do*-calculus in causal inference, we introduce **B**lack-box **B**ackdoor detection under the **Ca**usality **L**ens (BBCaL) which distinguishes backdoor and clean samples by analyzing prediction consistency after progressively constructing counterfactual samples. Theoretical analysis also sheds light on the effectiveness of the BBCaL. Extensive experiments on three benchmark datasets validate the effectiveness and efficiency of our method.

## 1 Introduction

Deep neural networks (DNNs) have achieved tremendous success in various applications (Kortli et al., 2020; Zou et al., 2023; Vaswani et al., 2017; Sun et al., 2022a; Zhou et al., 2024a;b). Despite these successes, training large DNNs requires considerable time and computational resources. Consequently, many users opt to utilize third-party pre-trained models through API requests (e.g., ChatGPT), or directly download them

from online platforms (e.g., ModelZoo, HuggingFace Hub). This setting is what we call Model-as-a-Service (MaaS) (Sun et al., 2022b; Li et al., 2017; Roman et al., 2009).

However, DNNs are easily attacked by injecting imperceptible backdoors during the training stage (Gu et al., 2017; Chen et al., 2017; Nguyen & Tran, 2021). After backdoor injection, the DNN's prediction results can be maliciously manipulated by the adversaries whenever the input sample contains the pre-defined trigger pattern, while it behaves normally when the input sample is benign. This vulnerability poses a serious threat to the MaaS setting, especially in some safety-critical applications such as autonomous driving (Han et al., 2022; Chan et al., 2022), medical diagnosis (Feng et al., 2022) and financial fraud detection (Lunghi et al., 2023).

To mitigate the threat in the MaaS setting, *inference-stage black-box backdoor detection* has drawn increasing research interest. In this scenario, defenders aim to establish a firewall-style detector on the user side to reject predictions for backdoored samples while forwarding predictions for clean samples. Several approaches have been proposed to address this issue (Guo et al., 2023; Liu et al., 2023). However, some crucial challenges still remain. ❶ **Strong Assumptions and Low Generalization:** Most defenses rely on strong assumptions about the trigger pattern (Gao et al., 2019; 2021; Guo et al., 2023) (e.g., sample-agnostic trigger) and are only feasible for specific types of attacks (Guo et al., 2023; Pal et al., 2024), which is impractical in MaaS scenarios where defenders lack information about attacks and triggers. ❷ **Low Efficiency:** Some detection algorithms (Liu et al., 2023) require extensive, computationally expensive image corruptions, leading to prolonged inference time that could degrade user experience. In light of these disadvantages, our objective is to *develop a generalized and efficient inference-stage detection algorithm. This algorithm allows defenders access only to the inference samples and their prediction labels generated by the DNNs, without any prior information about the target model and trigger patterns.*

To this end, we believe it is crucial to first address a fundamental question: *What is the inner mechanism that **causes** a backdoored DNN to behave normally with clean samples, but consistently predicts the target label when presented with backdoored samples?* To explore this, we introduce causal inference as a novel perspective to unravel the mechanisms behind the heterogeneous prediction behaviors of clean and backdoored samples through causal graphs. Specifically, our causal analysis derives that predictions of backdoored samples are primarily misled by the spurious path introduced by the attacks, whereas those of clean samples follow the original causal path. Notably, our causal analysis includes both sample-agnostic attacks, where triggers are identical across all samples, and sample-specific attacks, where triggers are customized for each sample. Analyzing this wide range of triggers through causal graphs enables us to develop a more generalized detection method.

Following these insights, a naïve detection method would be identifying the causal path that a sample follows during the inference stage. However, this approach is infeasible due to the limited information available. Therefore, motivated by causal interventions, which is used in causal inference to compare different potential outcomes under different variable interventions, we develop a **B**lack-box **B**ackdoor detection method under the **Ca**usality **L**ens (BBCaL). Specifically, we employ a simple yet effective counterfactual intervention method by progressively adding noise to the input sample. This approach implicitly induces distinguishable prediction behaviors between clean and backdoored samples, with theoretical analysis further elucidating its effectiveness. Extensive experiments demonstrate that our method can defend against a broader range of attacks with satisfactory efficiency.

In summary, the contributions of this paper include: **(1) Novel Causality Lens**. From the novel causality lens, we analyze the distinct prediction behaviors of clean images and various types of backdoored images in the MaaS setting. **(2) Counterfactual Backdoor Detection Algorithm**. Leveraging the causal analysis, we develop a causality-inspired detection method that distinguishes backdoor and clean samples by constructing counterfactual interventions and checking the consistency of the predictions. **(3) Theoretical Proof**. We provide a theoretical analysis (Theorem 8) to validate our method. **(4) Generaliztion and Efficieny**. Various experiments across multiple popular datasets have empirically proven that our detection method achieves an average of $> 84\%$ AUC under a wide range of attacks with only an ignorable overhead.

## 2 Related Work

**Backdoor Attacks**. Backdoor attacks are usually launched by poisoning training dataset. Specifically, malicious attackers inject trigger patterns into victim samples and alter the ground-truth label to a predefined target label. Recent research can be divided into two categories on making the backdoor attacks more stealthy to enhance the attacks' feasibility in practice. The first direction aims to make the trigger pattern visually less noticeable (Chen et al., 2017; Liu et al., 2020; Qi et al., 2022). For example, (Chen et al., 2017) blends the clean images with random pixels, and (Liu et al., 2020) uses the natural reflection to construct the backdoor trigger. The other direction aims to make the attacking process less noticeable (Shafahi et al., 2018; Souri et al., 2021; Zeng et al., 2022; Guan et al., 2024b; 2023b; Guo et al., 2024). For example, CL (Shafahi et al., 2018) proposes a clean-label attack, which constructs backdoor samples without changing labels.

**Backdoor Defenses**. To mitigate the threat of backdoor attacks, various defense methods have been proposed. As in (Li et al., 2022), we categorize the existing defense methods into five categories. First, detection-based defenses (Gao et al., 2019; Guo et al., 2021; Xiang et al., 2022; Guan et al., 2023a; Guo et al., 2023; Ma et al., 2022; Guan et al., 2024a) aim to explore whether the backdoors exist in the model. Second, preprocessing-based defenses (Doan et al., 2020; Shi et al., 2023) further introduce a preprocessing module so that triggers can be inactivated. Third, defenses based on model reconstruction (Liu et al., 2018; Zhao et al., 2020) directly eliminate the effect of backdoors by tuning model weights or adjusting model structures. Fourth, defenses based on trigger synthesis (Wang et al., 2019; Chen et al., 2022) first reverse engineer trigger patterns and then suppress the trigger's effects. Lastly, training-sample-filtering-based defenses (Li et al., 2021a; Huang et al., 2022a) work by first filtering backdoor samples from the training dataset, then training the model exclusively in the remaining dataset.

**Causal Inference in Backdoor Learning**. Causal inference (Yao et al., 2021; Hu et al., 2024; Chu et al., 2024) has been utilized as a tool to analyze attacks, defenses, and detection mechanisms in AI security (Huang et al., 2022b; Ren et al., 2022; Tople et al., 2020). For instance, (Ren et al., 2022) explores the causality between deep neural network outputs and subregions of input samples to uncover the working mechanism of adversarial examples. (Huang et al., 2022b) models the data generation process using structural causal models to distinguish between benign and malicious out-of-distribution (OOD) data. Additionally, (Tople et al., 2020) proposes using causal learning to mitigate privacy attacks. However, their causality analysis focuses on other well-established problems, such as OOD detection, adversarial example detection, and privacy attacks, which are fundamentally different from our backdoor detection problem. Although previous works (Zhang et al., 2023; Liu et al., 2024) have also explored the use of causal graphs in backdoor attacks, our causality analysis is fundamentally different from them: Their analysis aims to provide an analysis for **training a clean model from a backdoored dataset**, while our analysis aims to **investigate the distinct prediction behaviors in the inference stage** for clean and backdoored images. A more detailed comparison of the two papers is presented in Appendix O. To the best of our knowledge, this paper is the first to provide a causality analysis of backdoor attacks in the inference stage.

## 3 Preliminaries

### 3.1 Main Pipeline of Backdoor Attacks

Let $\mathcal{D} = \{\boldsymbol{x}_i, y_i\}_{i=1}^n$ denote the original dataset, where $\boldsymbol{x}_i \in \mathcal{R}^n$ denotes an image sample, and $y_i$ denotes the corresponding ground-truth label. The deployed neural network model is denoted as $f_\theta$, with $\theta$ as the trainable parameters. Then the malicious backdoor attacker selects a subset of the original dataset (denoted as $\mathcal{D}_c$) and modifies it to a backdoored version with $\mathcal{D}_b = \{(\hat{\boldsymbol{x}}_i, y_t) | \hat{\boldsymbol{x}}_i = \boldsymbol{x}_i + \delta(\boldsymbol{x_i}), (\boldsymbol{x}_i, y_i) \in \mathcal{D}_c\}$, where $y_t$ denotes the target label, and $\delta(\cdot)$ is a pre-defined trigger generation function. For example, in BadNet (Gu et al., 2017), the trigger generation function can be formulated as $\delta(\boldsymbol{x}_i) = (t - \boldsymbol{x}_i) \odot m$, where $t$ denotes the trigger, and $m$ is a binary mask defining the position of the trigger; in Blend (Chen et al., 2017), the trigger generation function can be formulated as $\delta(\boldsymbol{x}_i) = t$, where $t$ denotes the global trigger such as random noise. Following the taxonomy in (Li et al., 2022), we categorize backdoor attacks into "sample-agnostic" and "sample-specific" based on the properties of trigger function $\delta(\cdot)$ and provide the corresponding formal definitions as follows:

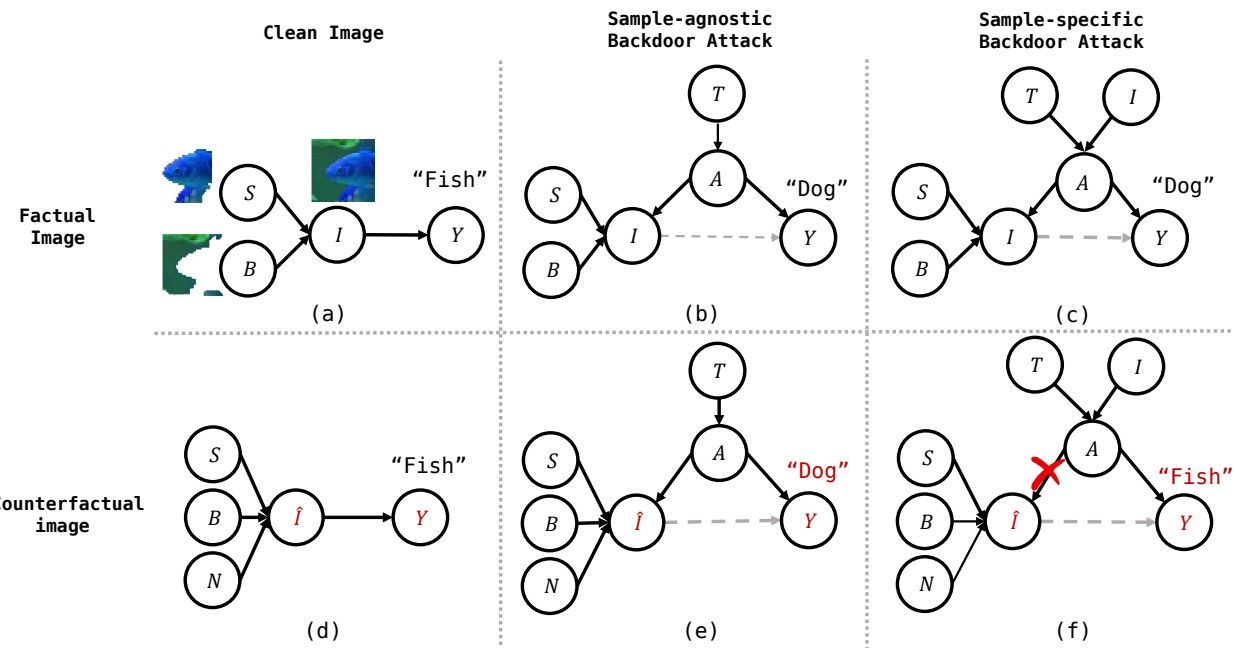

Figure 1: Causal graphs of the DNN's predictions for (a) clean images, (b) backdoor images with sample-agnostic trigger, (c) backdoor images with the sample-specific trigger, (d) counterfactual of clean images, (e) counterfactual of backdoor images with sample-agnostic trigger, and (f) counterfactual of backdoor images with the sample-specific trigger.

**Definition 1** (Sample-agnostic Trigger). *The trigger function $\delta(\boldsymbol{x}_i)$ of sample-agnostic backdoor attacks contain the same t, for all $\boldsymbol{x}_i$.*

**Definition 2** (Sample-specific Trigger). *The trigger function $\delta(\boldsymbol{x}_i)$ of sample-specific backdoor attacks contain different t for different $\boldsymbol{x}_i$.*

Then the neural network model $f_\theta$ is trained on the mixture of backdoored subset $\mathcal{D}_b$ and the remaining clean subset $\mathcal{D}_{/c}$ with the following optimization objective:

$$\min_\theta \sum_{i=1}^{|\mathcal{D}_b|} \ell(f_\theta(\hat{\boldsymbol{x}}_i), y_t) + \sum_{i=1}^{|\mathcal{D}_{/c}|} \ell(f_\theta(\boldsymbol{x}_i), y_i), \tag{1}$$

where $\ell(\cdot)$ denotes the loss function. In the inference stage, $f_\theta$ exhibits normal behavior when the input images are benign, but consistently predicts the target label $y_t$ when the trigger is present.

### 3.2 Problem Formulation

In this paper, we consider inference-stage backdoor detection under the black-box setting for the MaaS following (Guo et al., 2023; Liu et al., 2023). Specifically, the defender is assumed to only have access to input images and their prediction labels generated by the DNN, without any prior information about the backdoor attacks or the model. Our problem is hereby formally stated as follows:

**Problem 3** (Inference-stage Black-box Backdoor Detection). *Given a DNN $f_\theta(\cdot)$, the defender aims to build a detector $\mathcal{A}$ such that $\mathcal{A}(f_\theta(\boldsymbol{x}_{backdoor}), \boldsymbol{x}_{backdoor}) = 1$ if and only if the input is backdoored.*

The challenge of the problem 3 stems from two factors: (1) Generalization for various attacks: $\mathcal{A}$ aims to detect a wide range of backdoor attacks without prior knowledge of the target model and trigger patterns; (2) Efficiency guarantee: Our detection algorithm is expected not to significantly impact the inference efficiency.

### 3.3 Backdoor Attacks under a Causality Lens.

To unravel the complex problem 3, it is necessary to first answer a pivotal question: *What **causes** backdoored DNNs to consistently predict the target label for backdoored samples, yet behave normally for clean samples?* Motivated by causal inference (Xiao et al., 2023; Zhang et al., 2023), we propose using a novel **causal lens** to explore this question. Specifically, we construct causal graphs, which are directed acyclic graphs that illustrate the causal relationships between variables, for both clean and backdoored samples (see Figure 1). We provide a detailed analysis in the subsequent sections.

**Causal Graph for Clean Samples.** As shown in Figure 1(a), for a well-trained DNN, the prediction $Y$ of a clean image is dependent on the image content $I$, which consists of both semantic features $S$ and background features $B$. Thus, the causal relationship between $S$, $B$, and $I$ is represented as $(S \to I \leftarrow B)$. For instance, consider a fish image $I$, where pixels related to the fish per se are the semantic features $S$, and pixels related to the water and aquatic plants are background features $B$. A well-trained DNN makes prediction by leveraging both fish pixels and background pixels contained in $I$, denoted as $I \to Y$. This is referred to as the direct causal path.

For backdoor samples, the analysis becomes more complicated. As illustrated in Figure 1(b) and (c), backdoor attacks $A$ modify images $I$ by injecting triggers and altering image labels to the target label, represented as $I \leftarrow A \to Y$. This introduces a spurious path from $I$ to $Y$, which lies outside the direct causal path $(I \to Y)$. The attacks thereby serve as a **confounder**, which builds and strengthens the erroneous correlations between the modified images and the target label. Consequently, predictions of backdoored images are predominantly led by this spurious path $(I \leftarrow A \to Y)$ (Du et al., 2021; Zhang et al., 2023), while the direct causal path $(I \to Y)$ plays a minor role, symbolized by a gray dotted line in Figure 1 (a) and (c). This phenomenon has been empirically highlighted in previous papers (Cai et al., 2022; Yu et al., 2021; Sandoval-Segura et al., 2022). To develop a defense method with advanced generative capabilities across a wide range of backdoor attacks, we explore not only attacks based on sample-agnostic triggers but also thoes with sample-specific triggers (described in § 3.1). Specifically, we construct causal graphs for these two types of backdoor attacks as follows.

**Causal Graph for Sample-agnostic Backdoor Attacks.** Sample-agnostic backdoor attacks imply that all backdoor samples are constructed with a constant trigger pattern (see Definition 1). Therefore, valid sample-agnostic attacks $A$ in Figure 1(b) are solely dependent on the trigger patterns $T$ while independent of the image content, denoted as $T \to A$.

**Causal Graph for Sample-Specific Backdoor Attacks.** Sample-specific backdoor attacks imply that each backdoor sample is paired with a unique trigger, which is invalid for any other sample. Therefore, sample-specific attacks in Figure 1(c) depend on both the trigger patterns $T$ and the images $I$.

## 4 Methodology

In this section, according to the causal analysis above, we first introduce counterfactual intervention to distinguish clean and backdoored samples by analyzing prediction consistency after progressively constructing counterfactual samples. Causal analysis and theoretical proof are also provided to explain the mechanism of our defense method. Then, we introduce the detailed method to detect backdoor attacks in the second step.

### 4.1 Step 1: Designing and Conducting Counterfactual Intervention

The above analysis sheds light on the intuition that clean samples and backdoor samples can be distinguished by evaluating whether the model's predictions are primarily influenced by the direct causal path or the spurious path. However, the evaluations are challenging due to the black-box nature of DNNs and the limited information available in our setting. In causal inference, *do*-calculus is usually used to compare different potential outcomes under different counterfactual interventions on a specific variable. Specifically, *do*-calculus performs counterfactual interventions by changing the original values of one variable $t$ to a counterfactual value $t'$ (Guo et al., 2020; Yao et al., 2021). This method inspires us with the insight that, due to the intrinsically different causal rationales behind clean and backdoored samples, as demonstrated in the previous

causal graphs, when the same counterfactual intervention is applied to them, they may exhibit distinguishable prediction behaviors.

Then the main question is narrowed down to the following: *How to design an ideal counterfactual intervention strategy that increases the difference between clean and backdoored samples?* Traditional counterfactual intervention in causal inference (Guo et al., 2020; Yao et al., 2021) is not suitable for image settings because simply setting all images to a fixed value is meaningless and fails to induce different prediction behaviors between clean and backdoored samples. In addition, the detection method is expected to be highly efficient. Therefore, generating time-consuming counterfactual perturbations is impractical in the MaaS scenario. To design simple and effective counterfactual interventions, we propose to use random noise, which has been one of the most popular methods for conducting counterfactual interventions (Chang et al., 2018; Jeanneret et al., 2023). In addition, we also investigate other common counterfactual generation methods, such as random mask (Xiao et al., 2023) and mixup (Yu et al., 2023), in § 5.3. Empirically, random noise has proven to be the most stable and effective method for constructing counterfactual samples. Hence, we have adopted it in our method. We now formalize the idea and validate the idea with causal analysis and theoretical proof.

We first define a magnitude set $\mathbb{S}$, e.g., $\mathbb{S} = \{0.6, 0.2, 0.4, ..., 1.4\}$. Then, for each element $\alpha^j$ in $\mathbb{S}$, we construct a modified sample $\tilde{\boldsymbol{x}}_i^j = \boldsymbol{x}_i' + \alpha^j \cdot \boldsymbol{\epsilon}$, where $\boldsymbol{x}_i'$ denotes the input image, $\boldsymbol{n}_i = \alpha^j \cdot \boldsymbol{\epsilon}$ denotes the additive noise, $\alpha^j \in \mathbb{S}$ denotes the noise magnitude, and $\boldsymbol{\epsilon} \sim \mathcal{N}(0, \mathbb{I}_{\dim(\boldsymbol{x}_i')})$ denotes a random Gaussian noise. The following analysis illustrates why and how conducting interventions on clean samples and backdoored samples results in distinct prediction results after being fed into DNN.

**Prediction for Counterfactual Clean Images.** In Figure 1(d), upon introducing noise, the predictions of the counterfactual images are determined by the new images $\hat{I}$, which comprise the corresponding noise $N$, the original semantic features $S$, and the background features $B$. Specifically, the influence of the original semantic and the background features remains dominant when $a^j$ is small, leading to unchanged predictions ($f_\theta(\tilde{\boldsymbol{x}}_i^j) = y_i$). However, predictions change after introducing a sufficient amount of noise ($a^j$ is a medium value ). We validate this intuition with theoretical analysis in the later part.

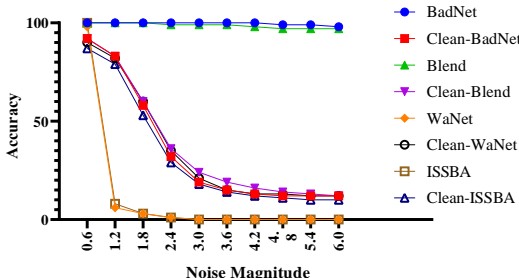

Figure 2: Accuracy of the backdoored DNNs on benign testset and backdoored testset under different noise magnitudes.

**Prediction for Counterfactual Backdoored Images.** For a backdoored image $\hat{\boldsymbol{x}}_i$, we treat the modified image $\tilde{\boldsymbol{x}}_i^j$ as a combination of a new image $\boldsymbol{x}_i''$ and the original backdoor trigger $\boldsymbol{t}_i$:

$$\tilde{\boldsymbol{x}}_i^j = \hat{\boldsymbol{x}}_i + n_j = \boldsymbol{x}_i + \delta(\boldsymbol{x}_i) + n_j = (\boldsymbol{x}_i + n_j) + \delta(\boldsymbol{x}_i) = \boldsymbol{x}_i'' + \delta(\boldsymbol{x}_i). \tag{2}$$

After adding noise to a backdoored image with **sample-agnostic triggers** (Figure 1(e)), the original valid trigger $\boldsymbol{t}_i$ remains effective for the new image $\boldsymbol{x}_i''$ due to a sample-agnostic trigger can poison any clean image. As a result, the prediction of the new backdoored image continues to be influenced primarily by the spurious path $\hat{I} \leftarrow A \rightarrow Y$. Consequently, the prediction remains unchanged until the image is significantly distorted by noise (e.g., for large $a^j$). For images with **sample-specific triggers**, as depicted in Figure 1(f), where triggers are tailored to individual images, the original backdoor triggers become ineffective for new images. As a result, the backdoor path ($A \rightarrow \hat{I}$) for new images is severed, with predictions mainly influenced by new images $\hat{I}$. Consequently, the predictions of images promptly deviate from the original target label $y_t$ upon adding noise. The theoretical proof is provided in Appendix P.

**Remark 4.** *In summary, images with sample-specific triggers witness immediate prediction flipping upon introducing noise, whereas clean images experience gradual outcome changes in response to noise intensity. Images with sample-agnostic triggers, however, maintain stability even with considerable added noise.*

### 4.1.1 Preliminary Experiments

To further validate above analysis, we conducted a preliminary study to substantiate the phenomenon.

**Setting.** We consider four popular backdoor attacks, specifically BadNet (Gu et al., 2017), Blend (Chen et al., 2017), WaNet (Nguyen & Tran, 2021), and ISSBA (Li et al., 2021b), where BadNet and Blend correspond to sample-agnostic backdoor attacks, while WaNet and ISSBA correspond to sample-specific backdoor attacks. All experiments are conducted on CIFAR-10 using ResNet18 as the neural network architecture. To successfully inject backdoors into the target model, we train the neural network for 200 epochs to achieve a clean accuracy of at least 90% and an attack success rate of at least 98%. For each backdoored and benign testset, we record the classification accuracy at different magnitudes of noise and plot the resultant curves in Figure 2. Specifically, to calculate the prediction accuracy of counterfactual samples, we consistently use the original labels from both the benign and backdoored test sets as the ground truth.

**Observation.** Note that the x-axis denotes the noise magnitude and the y-axis denotes the accuracy performance. Experiments conducted on clean and poisoned datasets under various backdoor attacks are plotted using different markers. For example, the accuracy performance on the backdoored and benign testset under the model backdoored with BadNet are drawn in blue and red color, respectively. As shown in Figure 2, as the magnitude of the noise increases, the accuracy curves for the benign test set steadily decrease. In contrast, those for the backdoored testset fall into two categories: 1) for sample-agnostic backdoor attacks like BadNet and Blend, accuracy performance is largely unaffected by noise magnitude; 2) for sample-specific backdoor attacks like WaNet and ISSBA, accuracy performance shows a sharp decline with only a small increase in noise. This observation aligns with the intuition discussed in the previous section.

### 4.1.2 Theoretical Analysis

Furthermore, we employ the neural tangent kernel (NTK) (Jacot et al., 2018; Guo et al., 2023; 2021; Lee et al., 2019) as a theoretical framework to shed light on the above phenomenon. Under the assumptions of NTK, a deep neural network with infinite width can be approximated as a linearized network. Hence, We first introduce an important lemma of NTK that is key to our proof.

**Lemma 5** (Infinite width networks as linearized networks (Lee et al., 2019)). *Let $f(x)$ denote a fully-connected neural network with $L$ hidden layers, each with width $n_l$ for $l = 1, \ldots, L$. Let $f_t(x) = h^{L+1}(x) \in \mathbb{R}^k$ denote the output of the neural network at time $t$. For a neural network $f_t(x)$ with infinite width, let $\Theta$ denote the corresponding deterministic kernel, and $f_t^{lin}(x)$ denote the linearization of $f_t(x)$. If the learning rate $\eta$ satisfies $\eta < \eta_{critical} := 2 \left( \lambda_{min}(\Theta) + \lambda_{max}(\Theta) \right)^{-1}$, where $\lambda_{min/max}(\Theta)$ are the minimum and maximum eigenvalues of $\Theta$, respectively, then for every $x \in \mathbb{R}^{n_0}$ with $||x||_2 \leq 1$, as $n \to \infty$ and $t \to \infty$, $f_t(x)$ converges in distribution to the same Gaussian distribution as the linearized model $\Theta(\mathcal{X}_T, \mathcal{X})\Theta^{-1}\mathcal{Y}$.*

Following (Guo et al., 2021; 2023), we leverage the RBF kernel and treat the DNN as a $k$-way kernel least square classifier. Under Lemma 5, the regression solution for the NTK is as follows:

$$\phi_q^b(\cdot) = \frac{\sum_{i=1}^{N_b} \Theta(\cdot, \boldsymbol{x}_i) \cdot y_q}{\sum_{i=1}^{N_b} \Theta(\cdot, \boldsymbol{x}_i) + \sum_{j=1}^{N_p} \Theta(\cdot, \boldsymbol{x}_j)}, \tag{3}$$

$$\phi_t^p(\cdot) = \frac{\sum_{i=1}^{N_b} \Theta(\cdot, \boldsymbol{x}_i) \cdot y_t + \sum_{j=1}^{N_p} \Theta(\cdot, \boldsymbol{x}_j) \cdot y_t}{\sum_{i=1}^{N_b} \Theta(\cdot, \boldsymbol{x}_i) + \sum_{j=1}^{N_p} \Theta(\cdot, \boldsymbol{x}_j)}, \tag{4}$$

where $\phi_q^b(\cdot)$ and $\phi_t^p(\cdot) \in \mathbb{R}$ represent the predicted probabilities of class $q$ and class $t$ based on $f(\cdot; \theta)$ for clean samples and backdoored samples, respectively. $N_b$ and $N_p$ denote the number of clean and backdoored samples, respectively. $\boldsymbol{x}_i$ and $\boldsymbol{x}_{iq}$ represent benign samples and benign samples specifically from class $q$, respectively. $\boldsymbol{x}_j$ represents the poisoned samples. $y_q$ and $y_t$ are the corresponding one-hot labels for class $q$ and target class $t$, respectively. The kernel function $\Theta(\boldsymbol{x}, \boldsymbol{x}_i) = e^{-2\gamma||\boldsymbol{x}-\boldsymbol{x}_i||^2}$ ($\gamma > 0$) is used. Given the assumption that training samples are evenly distributed, there are $\frac{N_b}{k}$ benign samples belonging to each class. Without loss of generality, we assume $y_q = 1$ for class $q$ while others are set to 0. The regression solution is then expressed as:

$$\phi_q^b(\cdot) = \frac{\sum_{i=1}^{N_b/k} \Theta(\cdot, \boldsymbol{x}_{iq})}{\sum_{i=1}^{N_b} \Theta(\cdot, \boldsymbol{x}_i) + \sum_{j=1}^{N_p} \Theta(\cdot, \boldsymbol{x}_j)}, \tag{5}$$

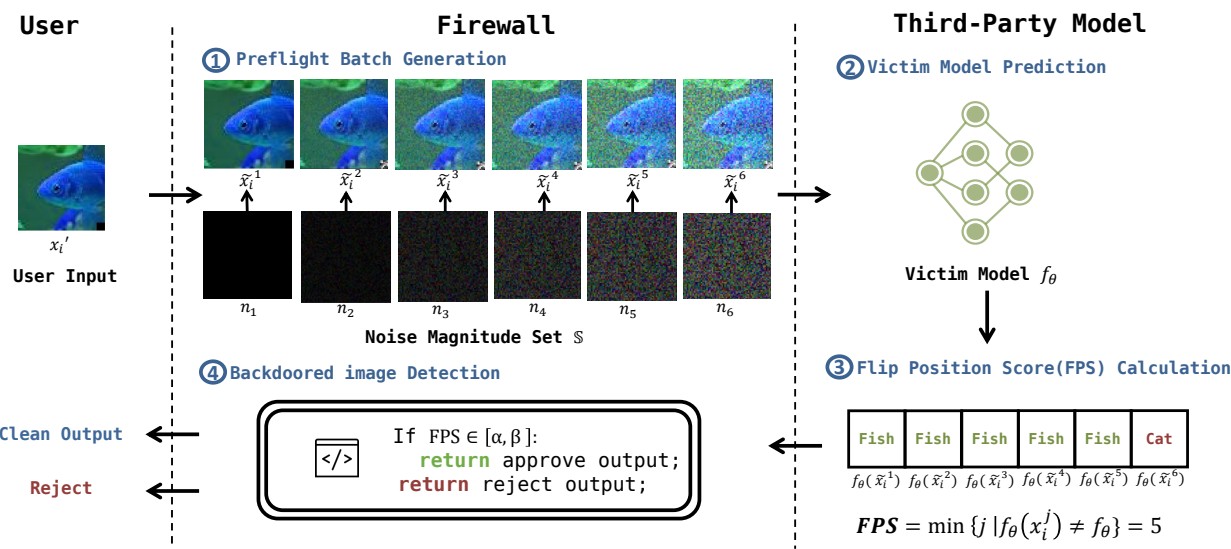

Figure 3: Pipeline of BBCaL. Upon receiving a target image $x'_i$ from the user, BBCaL first generates a counterfactual 'preflight batch' by incrementally introducing noise to $x'_i$. These counterfactual images are then fed to the DNN $f_\theta(\cdot)$ to obtain predictions. Subsequently, the FPS score is computed based on these predictions. Finally, BBCaL employs the FPS score to discern and decline queries from identified backdoored images while approving and outputting predictions for clean images.

$$\phi^p_t(\cdot) = \frac{\sum_{i=1}^{N_b/k} \Theta(\cdot, \boldsymbol{x}_{it}) + \sum_{j=1}^{N_p} \Theta(\cdot, \boldsymbol{x}_j)}{\sum_{i=1}^{N_b} \Theta(\cdot, \boldsymbol{x}_i) + \sum_{j=1}^{N_p} \Theta(\cdot, \boldsymbol{x}_j)}. \tag{6}$$

**Theorem 6** (Prediction consistency for clean samples). *Let $N_b$ denote the total number of benign samples, and let $\frac{N_b}{k}$ represent the number of benign samples in a specific class. Given a well-trained backdoored model $f_\theta$ and a clean input $\boldsymbol{x}$ from class $q$, define $\alpha = \mathbb{E}[||\boldsymbol{x} - \boldsymbol{x}_q||^2]$ and $\beta = \mathbb{E}[||\boldsymbol{x} - \boldsymbol{x}_{/q}||^2]$, where $\boldsymbol{x}_q$ and $\boldsymbol{x}_{/q}$ denote benign samples from class $q$ and benign samples from other class, respectively. Assuming that $\boldsymbol{x}_{/q}$ follows a Gaussian distribution, if we perturb the clean input with scaled Gaussian noise $\boldsymbol{\epsilon} \sim \mathcal{N}(0, a_j I)$, the prediction remains unchanged until $\boldsymbol{\epsilon} > \frac{\alpha^2 + \frac{\ln m}{2\gamma} - \beta^2}{2(\beta - \alpha)}$.*

*Proof.* The proof is provided under Lemma 5 and detailed in Appendix P. $\square$

The above theorem states that if the introduced noise $\boldsymbol{\epsilon}$ is small, the prediction for clean samples remains unchanged until $\boldsymbol{\epsilon} > \frac{\alpha^2 + \frac{\ln m}{2\gamma} - \beta^2}{2(\beta - \alpha)}$. This result is consistent with our causal analysis. Due to space constraints, the theoretical analysis for backdoored samples is provided in Appendix P. Hence, we can distinguish clean samples and backdoored samples by the different thresholds of prediction flipping in terms of counterfactual samples.

## 4.2 Step 2: Detecting Backdoor Samples

With the proposed counterfactual analysis, a straightforward method for detecting backdoored samples is as follows: We can progressively add random noise to the input image and use the minimum noise magnitude that flips the prediction results to determine whether the input is clean or not.

Formally, given a DNN $f_\theta(\cdot)$ and a test sample $\boldsymbol{x}_i$, we obtain a batch of counterfactual samples by progressively adding noise to the test sample, denoted as the "preflight batch", consisting of modified images: $P_i =$

$\left[\tilde{\boldsymbol{x}}_i^1, \tilde{\boldsymbol{x}}_i^2, ..., \tilde{\boldsymbol{x}}_i^{|\mathbb{S}|}\right]$. After querying the deployed DNN with the preflight batch, we record the corresponding prediction results for the sample $\boldsymbol{x}_i$: $R_i = \left[f_\theta(\tilde{\boldsymbol{x}}_i^1), f_\theta(\tilde{\boldsymbol{x}}_i^2), ..., f_\theta(\tilde{\boldsymbol{x}}_i^{|\mathbb{S}|})\right]$.

As discussed in § 4.1, the position of prediction flipping differs in clean and backdoored images. To exploit this difference, we introduce a novel metric, the Flip Position Score (FPS), to measure the prediction changing time (detailed algorithm in Appendix C). For each sample, $\boldsymbol{x}_i$, the FPS is defined as

$$FPS(\boldsymbol{x}_i) = \min\{j | f_\theta(\tilde{\boldsymbol{x}}_i^j) \neq f_\theta(\tilde{\boldsymbol{x}}_i^1)\}. \tag{7}$$

In essence, FPS computes the index that the prediction result $f_\theta(\tilde{\boldsymbol{x}}_i^j)$ differs from the initial result $f_\theta(\tilde{\boldsymbol{x}}_i^1)$. Backdoored images typically have extremely low or high FPS scores, whereas clean images often have scores centered around the median. Hence, if $FPS(\boldsymbol{x}_i)$ lies within the threshold range $[\alpha, \beta]$, the sample is classified as clean; otherwise, it is considered a backdoored sample. An overview of this approach is provided in Algorithm 2 and visualized in Figure 3. Details about the magnitude set and threshold range are elaborated in the experimental section.

## 5 Experiments

### 5.1 Experimental Settings

**Datasets and Models.** Following (Guo et al., 2023; Gao et al., 2019; Li et al., 2021a), we choose three popular datasets for evaluating the effectiveness of our proposed method: CIFAR-10 (Krizhevsky, 2009), GTSRB (Stallkamp et al., 2012), and ImageNet-subset (git). The details of the three datasets are listed in Table 3. For CIFAR-10 and GTSRB, we train with the popular ResNet (He et al., 2015). However, for the ImageNet-subset, we opted for the EfficientNet architecture (Tan & Le, 2020) as it reports a higher accuracy.

**Attack Baselines.** We choose 9 backdoor attacks from the well-established recent works as our baselines: 1) BadNet (Gu et al., 2017), 2) Blend Attack (Chen et al., 2017), 3) Label-Clean (Shafahi et al., 2018), 4) SIG (Barni et al., 2019), 5) WaNet (Nguyen & Tran, 2021), 6) ISSBA (Li et al., 2021b), 7) Adaptive Blend (Qi et al., 2022), 8) Filter (Liu et al., 2019), and 9) DFST (Cheng et al., 2021). All attack baselines are implemented with the open-sourced backdoor learning toolbox (Li et al., 2023). More details of each attack method can be found in Appendix F.

**Defense Baselines.** Based on our setting, it is assumed that defenders can only access the prediction results and the input images. Therefore, we compare our method with ScaleUP (Guo et al., 2023), which perfectly fits the setting. In addition, we compare our method with Frequency (Zeng et al., 2021) and LAVA (Just et al., 2023), which require an additional validation set, and STRIP (Gao et al., 2019), which requires additional prediction probability information from the DNN model. More details about the defense baselines are in Appendix G.

**Implementation Details.** Following the previous works in backdoor defense (Li et al., 2021a), the poisoning ratio for backdoor attacks is set as 10% as default. The length of the magnitude set $\mathcal{S}$ has been set to 7 based on the experiments described in Section 5.3. According to our previous analysis, *alpha* should be small to detect sample-specific backdoor attacks, while $\beta$ should be large to identify sample-agnostic backdoor attacks. Hence, the values of $\alpha$ and $\beta$ are set as 1 and 6, respectively, which represent the first position and the second-to-last position in the magnitude set. All the hyperparameters are evaluated with ablation studies in the later part. Full details of the implementation are provided in Appendix J.

**Evaluation Metrics.** Following the existing works in backdoor detection (Gao et al., 2021; Guo et al., 2021; 2023), we choose the precision (P), recall (R), and the area under receiver operating curve (AUROC) as the evaluation metrics.

### 5.2 Main Results

Table 1 presents the main result, where we compare our method with other defense baselines against various backdoor attacks on three datasets. For each metric, we mark the highest value with the **bold** font and the

Table 1: Comparison of the proposed method with other baseline defense methods in terms of precision(P), Recall(R), and AUROC(AUC).

| Dataset | Attack Method ↓ | STRIP [1] | | | Frequency [2] | | | LAVA [2] | | | Scale-UP | | | Ours | | |
|---|---|---|---|---|---|---|---|---|---|---|---|---|---|---|---|---|
| | | P | R | AUC | P | R | AUC | P | R | AUC | P | R | AUC | P | R | AUC |
| CIFAR-10 | BadNet | 0.85 | 0.99 | 0.98 | 0.76 | 0.64 | 0.82 | 0.70 | 0.57 | 0.78 | **0.88** | **1.00** | **0.93** | 0.85 | **1.00** | 0.91 |
| | Blend | 0.81 | 0.82 | 0.89 | 0.82 | 0.89 | 0.90 | 0.51 | 0.45 | 0.67 | 0.55 | 0.34 | 0.56 | **0.83** | **0.97** | **0.90** |
| | WaNet | 0.40 | 0.10 | 0.46 | 0.16 | 0.04 | 0.27 | 0.36 | 0.01 | 0.71 | 0.74 | 0.80 | 0.78 | **0.81** | **0.90** | **0.95** |
| | ISSBA | 0.55 | 0.49 | 0.49 | 0.55 | 0.25 | 0.56 | 0.53 | 0.01 | 0.77 | 0.79 | 0.99 | 0.92 | **0.90** | **1.00** | **0.95** |
| | Label-Clean | **0.89** | 0.45 | 0.91 | 0.67 | 0.61 | 0.74 | 0.50 | **0.95** | 0.85 | 0.76 | 0.96 | 0.91 | 0.82 | **0.97** | **0.96** |
| | SIG | 0.52 | 0.99 | 0.84 | 0.56 | 0.60 | 0.85 | 0.43 | 0.50 | 0.58 | 0.32 | 0.45 | 0.48 | **0.80** | **1.00** | **0.90** |
| | Adaptive Blend | 0.50 | **1.00** | 0.94 | 0.45 | 0.46 | 0.64 | 0.50 | **1.00** | 0.50 | 0.58 | 0.74 | 0.62 | **0.81** | 0.73 | **0.96** |
| | Filter | 0.48 | 0.35 | 0.54 | 0.50 | **0.99** | 0.50 | 0.44 | 0.68 | 0.70 | 0.49 | 0.65 | 0.51 | **0.64** | **0.99** | **0.80** |
| | DFST | 0.51 | 0.99 | 0.70 | 0.50 | **1.00** | 0.45 | 0.67 | 0.54 | 0.72 | 0.46 | 0.55 | 0.57 | **0.78** | 0.99 | **0.90** |
| | **Average** | 0.61 | 0.68 | 0.75 | 0.55 | 0.61 | 0.64 | 0.52 | 0.52 | 0.70 | 0.62 | 0.72 | 0.70 | **0.80** | **0.95** | **0.91** |
| GTSRB | BadNet | 0.50 | 0.98 | 0.83 | 0.53 | 0.98 | **0.95** | 0.60 | 0.46 | 0.64 | 0.66 | **0.99** | 0.87 | **0.74** | **0.99** | 0.85 |
| | Blend | 0.56 | 0.99 | 0.84 | 0.53 | **0.99** | **0.97** | 0.72 | 0.68 | 0.76 | 0.56 | 0.74 | 0.59 | **0.76** | **0.99** | 0.87 |
| | WaNet | 0.53 | 0.94 | 0.75 | 0.52 | 0.92 | 0.65 | 0.63 | 0.45 | 0.67 | 0.74 | 0.87 | 0.86 | **0.98** | **0.93** | **0.91** |
| | ISSBA | 0.50 | 0.97 | 0.72 | 0.53 | **0.99** | 0.91 | 0.54 | 0.40 | 0.58 | 0.63 | 0.89 | 0.78 | **0.75** | 0.95 | **0.93** |
| | Label-Clean | **0.61** | 0.48 | 0.68 | 0.48 | 0.84 | 0.60 | 0.52 | **0.90** | 0.68 | 0.49 | 0.58 | 0.51 | 0.51 | 0.68 | **0.76** |
| | SIG | 0.49 | 0.68 | 0.50 | 0.50 | **0.94** | 0.60 | 0.74 | 0.75 | **0.80** | 0.49 | 0.69 | 0.50 | **0.83** | 0.92 | **0.90** |
| | Adaptive Blend | 0.50 | **1.00** | **0.75** | 0.52 | 0.94 | 0.58 | 0.52 | 0.57 | 0.64 | 0.40 | 0.51 | 0.65 | **0.67** | 0.80 | 0.73 |
| | Filter | 0.50 | **1.00** | 0.25 | 0.50 | 0.48 | 0.34 | 0.61 | 0.58 | 0.55 | 0.46 | 0.63 | 0.56 | **0.73** | 0.78 | **0.80** |
| | DFST | 0.50 | 0.99 | **0.88** | 0.50 | **1.00** | 0.32 | 0.65 | 0.63 | 0.70 | 0.57 | 0.98 | 0.75 | **0.72** | **1.00** | 0.82 |
| | **Average** | 0.52 | 0.89 | 0.69 | 0.51 | **0.90** | 0.66 | 0.61 | 0.60 | 0.67 | 0.56 | 0.76 | 0.67 | **0.74** | 0.89 | **0.84** |
| ImageNet-subset | BadNet | **0.90** | 0.67 | 0.92 | 0.77 | 0.92 | 0.84 | 0.65 | 0.47 | 0.74 | 0.50 | 0.29 | 0.60 | 0.87 | **0.98** | **0.93** |
| | Blend | 0.51 | **1.00** | 0.69 | **0.78** | 0.91 | **0.87** | 0.64 | 0.55 | 0.68 | 0.64 | 0.43 | 0.70 | 0.67 | 0.77 | 0.83 |
| | WaNet | 0.51 | 0.96 | 0.55 | 0.63 | 0.52 | 0.65 | 0.59 | 0.49 | 0.72 | 0.74 | 0.81 | 0.82 | **0.88** | **0.97** | **0.93** |
| | ISSBA | 0.57 | 0.92 | 0.60 | 0.76 | 0.42 | 0.62 | 0.58 | 0.56 | 0.55 | 0.70 | 0.87 | 0.81 | **0.89** | **0.95** | **0.94** |
| | Label-Clean | **0.63** | 0.48 | 0.64 | 0.52 | 0.79 | 0.54 | 0.46 | **0.85** | 0.62 | 0.51 | 0.53 | 0.52 | 0.62 | 0.73 | **0.80** |
| | SIG | **0.73** | **0.91** | **0.88** | 0.66 | 0.68 | 0.72 | 0.68 | 0.56 | 0.66 | 0.46 | 0.65 | 0.55 | 0.67 | 0.55 | 0.75 |
| | Adaptive Blend | 0.49 | **1.00** | 0.73 | 0.63 | **1.00** | 0.57 | 0.54 | 0.56 | 0.63 | 0.48 | 0.53 | 0.62 | **0.75** | 0.87 | **0.80** |
| | Filter | 0.54 | **0.97** | 0.46 | 0.61 | 0.59 | 0.60 | 0.44 | 0.65 | 0.67 | 0.49 | 0.70 | 0.51 | **0.69** | 0.75 | **0.77** |
| | DFST | 0.52 | **0.93** | 0.51 | 0.63 | 0.57 | 0.62 | 0.48 | 0.67 | 0.68 | 0.55 | 0.73 | 0.64 | **0.73** | 0.78 | **0.82** |
| | **Average** | 0.60 | **0.87** | 0.66 | 0.65 | 0.71 | 0.67 | 0.56 | 0.60 | 0.56 | 0.56 | 0.56 | 0.64 | **0.75** | 0.81 | **0.84** |

[1] STRIP requires additional prediction probability information.

[2] Frequency and LAVA both require an additional validation set.

second highest value with an underline. As the table suggests, our method achieves a promising performance on all three datasets against various attack methods. Especially for sample-specific backdoor attacks (e.g., WaNet and ISSBA), our method has been shown to be significantly better than the baseline defenses. Note that in our defense baselines, STRIP requires additional prediction probability information from the DNN model to detect backdoored samples, while our method only depends on the prediction labels. Moreover, Frequency and LAVA leverage an additional validation set, which is also not required by our method. Our method has performed on par or even surpassed these three baselines with less information about the DNN model and the dataset. SCALE-UP is developed for the same setting as our method. However, it shows a much worse performance when defending against the Blend attack. This can be attributed to the Blend attack utilizing a global trigger (e.g., Hello-kitty-like image), while the scaling operation in the SCALE-UP can easily destroy the feature information contained in the global trigger pattern. It is also noted that our method is seemingly less effective on Filter and DFST, compared to our promising performance on Badnet and Blend. We provide further discussion on this observation in the Appendix I.

## 5.3 Ablation Studies

**The impact of the poisoning ratio and trigger size.** To evaluate the effectiveness of BBCaL against different levels of poisoning ratio, we present the experimental results on CIFAR-10 in Figure 4. The results indicate that our method generally achieves stable performance across various levels of poisoning. Furthermore, Figure 7 shows that the performance of BBCaL remains stable across various trigger sizes, where the x-axis denotes the ratio of trigger size to image size in terms of length unit.

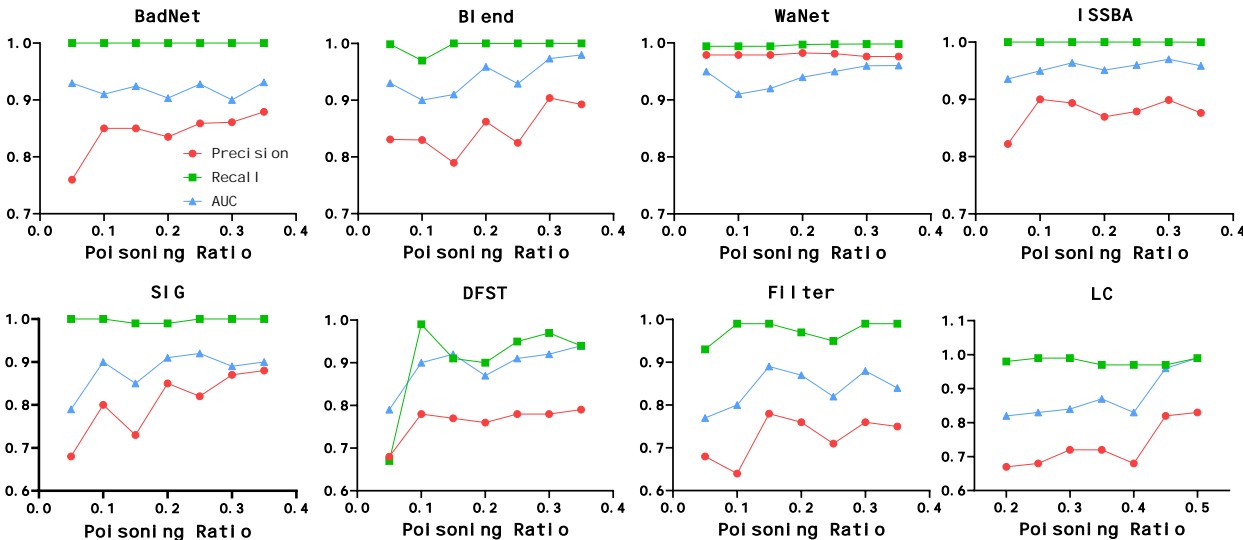

Figure 4: Performance with different poisoning ratios.

Table 2: Comparison between other common counterfactual generation methods

| Method | BadNet | | | Blend | | | WaNet | | | ISSBA | | | Label-Clean | | | SIG | | | Adaptive Blend | | | Filter | | | DFST | | |
|---|---|---|---|---|---|---|---|---|---|---|---|---|---|---|---|---|---|---|---|---|---|---|---|---|---|---|---|
| | P | R | AUROC | P | R | AUROC | P | R | AUROC | P | R | AUROC | P | R | AUROC | P | R | AUROC | P | R | AUROC | P | R | AUROC | P | R | AUROC |
| mix-up | 0.50 | **1.00** | **0.98** | 0.52 | 1.00 | **0.99** | 0.49 | **0.98** | 0.80 | 0.50 | 0.99 | 0.76 | **0.85** | 0.90 | 0.90 | 0.45 | 0.78 | 0.65 | 0.79 | **0.80** | 0.82 | 0.51 | 0.85 | 0.72 | 0.51 | **1.00** | 0.57 |
| mask | 0.57 | **1.00** | 0.89 | 0.55 | **1.00** | 0.88 | 0.53 | 0.81 | 0.47 | 0.54 | 0.89 | 0.83 | 0.76 | 0.96 | 0.87 | 0.56 | 0.60 | 0.56 | 0.78 | 0.76 | 0.82 | 0.48 | 0.64 | 0.62 | 0.68 | 0.62 | 0.71 |
| **noise** | **0.85** | **1.00** | 0.91 | **0.83** | 0.97 | 0.90 | **0.81** | 0.90 | **0.95** | **0.90** | **1.00** | **0.95** | 0.82 | **0.97** | **0.96** | **0.80** | **1.00** | **0.90** | **0.81** | 0.73 | **0.96** | **0.64** | **0.99** | **0.80** | **0.78** | 0.99 | **0.90** |

**The impact of counterfactual generation method.** Apart from random noise, other counterfactual generation methods, such as random masking (Xiao et al., 2023) and mixup (Yu et al., 2023), have also been widely used to generate counterfactual samples. To assess the impact of counterfactual generation design, we compare the performance of the default random noise strategy with the mixup and the random masking strategy on CIFAR-10 dataset and report the results in Table 2. As the table suggests, random noise shows the most stable and satisfactory performance across all types of backdoor attacks. A possible explanation is that random noise enables the generation of counterfactual samples with finer granularity.

**The impact of magnitude set.** Figure 5 and Figure 11 presents a series of heatmaps, showing the performance of BBCaL under different combinations of *length* and *step* against different backdoor attacks. In particular, *length* represents the length of the magnitude set $\mathbb{S}$, and *step* represents the difference between two adjacent noise values in the magnitude set. As shown, the performance of our method generally achieves satisfactory performance with a *length* of 7 and a *step* of 0.2.

### 5.4 Discussion and Visualization

**Efficiency Testing.** Efficiency is of critical concern in our setting, since user experience is expected to not be significantly affected by the detection algorithm. Therefore, we compare the inference time consumption

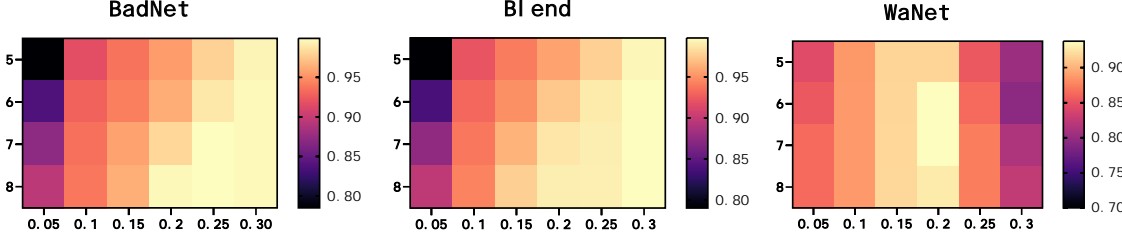

Figure 5: Performance of BBCaL under different combinations of *length* and *step*.

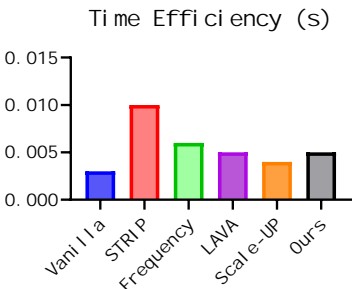

Figure 6: Comparison of inference time.

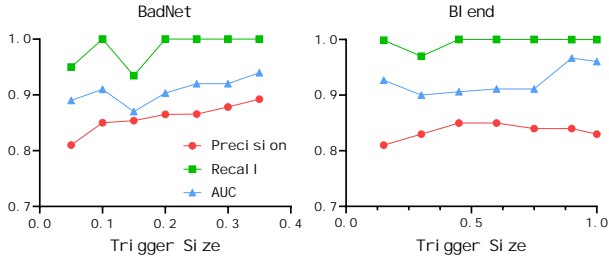

Figure 7: Performance against backdoor attacks with different trigger size.

before and after adopting the detection algorithms and report the corresponding results in Figure 6. The results demonstrate that our BBCaL ranks as the top-2 most efficient algorithm, exhibiting a trivial overhead compared to the vanilla inference time consumption (without detection algorithm), which proves the efficiency of our method.

**Score Distribution.** To visually demonstrate the effectiveness of BBCaL, we plot the distribution of the FPS values for backdoored samples and clean samples in Appendix L. As the figure suggests, the FPS values for clean samples are centered in the middle, but those for backdoored samples lie on the two sides, aligning with the causal analysis derived in the last section.

**Defences against Adaptive Attacks.** We consider a practical scenario where an attacker has prior information about our deployed defense method. Then, instead of training backdoored models as in Equation 1, the attacker might add an additional term to the loss function to evade our backdoor detection. Specifically, the adaptive loss function would be,

$$\min_{\theta} \sum_{i=1}^{|\mathcal{D}_{/c}|} \ell(f_{\theta}(\boldsymbol{x}_i), y_i) + \sum_{i=1}^{|\mathcal{D}_b|} \ell(f_{\theta}(\hat{\boldsymbol{x}}_i), y_t) + \sum_{i=1}^{|\mathcal{D}_b|} \ell(f_{\theta}(\hat{\boldsymbol{x}}_i + m \cdot \boldsymbol{\epsilon_i}), y_i), \tag{8}$$

where $\boldsymbol{\epsilon_i}$ is a random Gaussian noise added to each backdoor sample $\hat{\boldsymbol{x}}_i$, and $m$ is the magnitude multiplier of the added noise. Intuitively, this loss function makes the prediction behaviors on backdoored images more similar to those on clean images when the noise is added. To evaluate the effectiveness of our detection method, we provide results with varying $m$ against BadNet attack on CIFAR-10 dataset on Figure 8. As the figure suggests, the performance is consistently satisfactory with $m$ varying from 0.1 to 0.7.

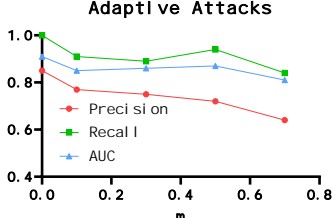

Figure 8: Performance against adaptive attacks with varying $m$.

## 6 Conclusion and Future Works

In this paper, we propose an effective method for solving the input-level black-box backdoor detection problem. Our method is firstly motivated by a novel perspective for analyzing the heterogeneous prediction behaviors for backdoored samples and clean samples, and further supported by a strict theoretical analysis. Then by leveraging the causal insight, our detection algorithm introduces counterfactual samples as an intervention in the prediction behaviors to distinguish backdoored samples and clean samples. Extensive experiments across popular datasets demonstrate the effectiveness and efficiency of our method. Despite these satisfactory results, there are still several directions that can be explored in the future. Firstly, how to construct an automatic algorithm for determining the magnitude set $\mathbb{S}$? Meta information such as the mean and variance of the training dataset might be helpful in the algorithm design. Secondly, we focus solely on classification task in the experiment. It would be promising if the methodology can be also adapted to other tasks such as generation.

## Acknowledgement

The work is in part supported by the U.S. Office of Naval Research Award under Grant Number N00014-24-1-2668 and the National Science Foundation under Grants IIS-2316306 and CNS-2330215.

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

## A   Appendix

## B   Preliminary Experiment on Counterfactual Examples

## C   FPS score

---

**Algorithm 1** FPS Score Calculation.

---

**Input:** Prediction results $f_\theta(P_i) = \left[ f_\theta(\tilde{x}_1^i), f_\theta(\tilde{x}_2^i), ..., f_\theta(\tilde{x}_{|\mathbb{S}|}^i) \right]$

FPS $= |\mathbb{S}|$
**for** $j = 1$ **to** $|\mathbb{S}|$ **do**
   **if** $f_\theta(\tilde{x}_j^i) \neq f_\theta(\tilde{x}_1^i)$ **then**
      **break**
   **else**
      FPS = j
   **end if**
**end for**
**return** FPS

---

## D   Algorithm

---

**Algorithm 2** The Backdoor Detection Method.

---

**Input:** Dataset $\mathcal{D}_{test} = \{(\boldsymbol{x}_1, y_1), ..., (\boldsymbol{x}_n, y_n)\}$; Target Model $f_\theta$; detection threshold $[\alpha, \beta]$; magnitude set $\mathbb{S}$.

**for** $i = 1$ **to** $n$ **do**
   Construct the preflight batch $P_i = \left[ \tilde{\boldsymbol{x}}_i^1, \tilde{\boldsymbol{x}}_i^2, ..., \tilde{\boldsymbol{x}}_i^{|\mathbb{S}|} \right]$ for each $(\boldsymbol{x}_i, y_i) \in \mathcal{D}_{test}$ given $\mathbb{S}$.

   Obtain the prediction result for the query $P_i$, $\left[ f_\theta(\tilde{x}_1^i), f_\theta(\tilde{x}_2^i), ..., f_\theta(\tilde{x}_{|\mathbb{S}|}^i) \right]$.
   Compute the score on $\boldsymbol{x}_i$ following Algorithm 1 in Appendix C.
**end for**
Filter backdoored samples by threshold $[\alpha, \beta]$.

---

## E   More Details about Dataset

The details of the dataset are given in Table 3.

Table 3: Statistical information about the Datasets

| Dataset | Image Size | # of Training samples | # of Testing Samples | # of Classes |
|---|---|---|---|---|
| CIFAR-10 | $32 \times 32 \times 3$ | 50,000 | 10,000 | 10 |
| GTSRB | $32 \times 32 \times 3$ | 39,209 | 12,630 | 43 |
| ImageNet-Subset | $224 \times 224 \times 3$ | 9,469 | 3,925 | 10 |

## F   More Details about Attack baselines

All the attack baselines are implemented with the open-sourced backdoor learning toolbox (Li et al., 2023). The details of the attack baselines are as below:

- **BadNet** (Gu et al., 2017) employs grid-like pixels as the triggers for each of the backdoored samples.

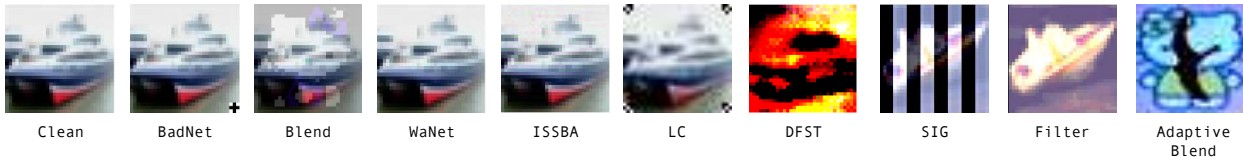

Figure 9: Visualization of backdoor samples on CIFAR-10 dataset.

- **Blend** (Chen et al., 2017) employs a hello-kitty-like image and blends it with each of the backdoored samples.

- **WaNet** (Nguyen & Tran, 2021) employs the interpolation method and generates sample-specific triggers for each of the backdoored samples.

- **ISSBA** (Li et al., 2021b) generates sample-specific trigger patterns through an encoder-decoder network.

- **LC** (Shafahi et al., 2018) employs perturbations to modify the original images and then conducts invisible backdoor attacks.

- **DFST** (Cheng et al., 2021) conducts the invisible backdoor attacks in the feature space via style transfer. They first train a style-transfer GAN model on sunset weather datasets and then employ the trained GAN model to construct backdoor images so that the backdoor images have a style of sunset.

- **SIG** (Barni et al., 2019) conducts the invisible backdoor attacks with sinusoidal signal.

- **Filter** (Liu et al., 2019) constructs backdoor images by employing filters such as the Nashville filter as the trigger pattern.

- **Adaptive Blend** (Qi et al., 2022) constructs backdoor samples by first partitioning the standard trigger patterns into several patches in the training stage. But in the inference stage, backdoors are activated with the standard trigger pattern.

We present a visualization of the backdoored image generated by different backdoor attacks in Figure 9.

## G More Details about Defense baselines

In this section, we introduce the basic parameter setting for each of the defense baselines.

- **STRIP** (Gao et al., 2019): We follow the official implementation of STRIP[1]. Specifically, 100 samples are iteratively superimposed on the given sample and we record the classification probabilities generated by the DNN model. Subsequently, an entropy value is calculated based on the 100 probability values to determine whether the given sample is a backdoor or not. A higher entropy value denotes a higher probability of being backdoored.

- **Frequency** (Zeng et al., 2021): We follow the official implementation of Frequency[2]. Specifically, we employ a 6-layer CNN model as the backbone architecture of the binary detector and train it for 50 epochs on an additional validation set of 1000 samples. The binary detector determines whether the given sample is clean or backdoored by analyzing the Fourier transform of the original image. Subsequently, we use the probability of being identified as "backdoored" as the score for each sample.

- **LAVA** (Just et al., 2023): We follow the official implementation of LAVA[3]. Specifically, we determine the data valuation of each data sample in the test set by calculating the proposed calibrated gradient in the original paper. A higher gradient value denotes a higher probability of being backdoored.

---

[1] https://github.com/garrisongys/STRIP
[2] https://github.com/YiZeng623/frequency-backdoor
[3] https://github.com/ruoxi-jia-group/LAVA

Table 4: Details about the deployed DNN models

| Dataset ↓ | Attack Method | | | | | | | | | | | | | | | |
|---|---|---|---|---|---|---|---|---|---|---|---|---|---|---|---|---|
| | BadNet | | Blend | | WaNet | | ISSBA | | SIG | | LC | | DFST | | Filter | | Adaptive Blend | |
| | CA (%) | ASR (%) | CA (%) | ASR (%) | CA (%) | ASR (%) | CA (%) | ASR (%) | CA (%) | ASR (%) | CA (%) | ASR (%) | CA (%) | ASR (%) | CA (%) | ASR (%) | CA (%) | ASR (%) |
| CIFAR-10 | 92 | 100 | 92 | 100 | 92 | 200 | 91 | 100 | 92 | 100 | 88 | 100 | 90 | 100 | 89 | 100 | 90 | 100 |
| GTSRB | 97 | 100 | 97 | 100 | 96 | 100 | 96 | 100 | 96 | 100 | 90 | 100 | 87 | 100 | 92 | 100 | 92 | 100 |
| ImageNet | 83 | 100 | 83 | 100 | 82 | 100 | 83 | 100 | 83 | 100 | 80 | 97 | 80 | 96 | 82 | 99 | 80 | 92 |

- **SCALE-UP** (Guo et al., 2023): We follow the official implementation of SCALE-UP [4]. Specifically, the scaling set is chosen as $S = \{1, 3, 5, 7, 9\}$ for all the experiments. The proposed SPC value is calculated for each of the samples, where a higher SPC value denotes a higher probability of being a backdoor sample.

## H  More Details about the Deployed DNN Models

We provide the details about the deployed DNN models in Table 4, where CA denotes the clean accuracy and ASR denotes the attack success rate. Note that for each experiment, we run the experiment three times and record the average performance of CA and ASR.

## I  Experimental Discussion

Our explanation is that, for semantic backdoor attacks such as Filter, and DFST, they all rely on semantic features as a trigger to launch backdoor attacks. For example, Filter relies on a specific filter, and DFST relies on a style transfer learned from a couple of sunset images. Therefore, when running our detection method over these backdoor attacks, it is observed that the distributions of these backdoor samples' FPS scores tend to be slightly closer to the clean samples' FPS scores. Specifically, (1) when we start adding random noise, the prediction label of these backdoor samples will not immediately change; (2) when we gradually add larger magnitudes of random noise, the prediction label of these backdoor samples will ultimately change with a faster speed than the clean samples, potentially because the semantic features in the trigger are not as complex as those in the clean image. Thereby, observation (1) makes the FPS not as low as the typical sample-specific backdoor attacks such as WaNet; and observation (2) makes the FPS not as stably high as the typical sample-agnostic backdoor attacks such as BadNets. Although it is slightly less effective, our method still outperforms the baselines. These experimental findings pose a noteworthy question that might be helpful for further refining our method. Inspired by this interesting question, we'd like to leave more explorations in the future works.

## J  More Details about Implementation

Following the prior works in backdoor defences (Li et al., 2021a), the poisoning ratio for backdoor attacks is set as 10% as default. The $\alpha$ and $\beta$ values are set as 1 and 6, respectively. It is noted that the design of the magnitude set $\mathbb{S}$ is a non-trivial question. If the set is too short, the granularity might not be fine-grained enough to distinguish between backdoor samples and clean samples, which is detrimental to meeting the effectiveness requirement. However, if the set is too large, the efficiency requirement cannot be satisfied. To achieve a balanced trade-off between the two sides, we have chosen 7 as a moderate length for the magnitude set, where the noise magnitude increases linearly starting from 0 with a step length of 0.2. For $\alpha$ and $\beta$, we choose 1 and 6 for all datasets, respectively. As shown in Figure 10, they are stable to distinguish backdoor and clean samples across all datasets. All the experiments are evaluated on an NVIDIA RTX A5000 GPU with 24GB GPU memory.

## K  Comparison of different counterfactual generation methods

**The impact of counterfactual generation method**  Apart from random noise, other methods, such as random masking (Xiao et al., 2023) and mixup (Yu et al., 2023), have also been extensively used for

---

[4]https://github.com/JunfengGo/SCALE-UP

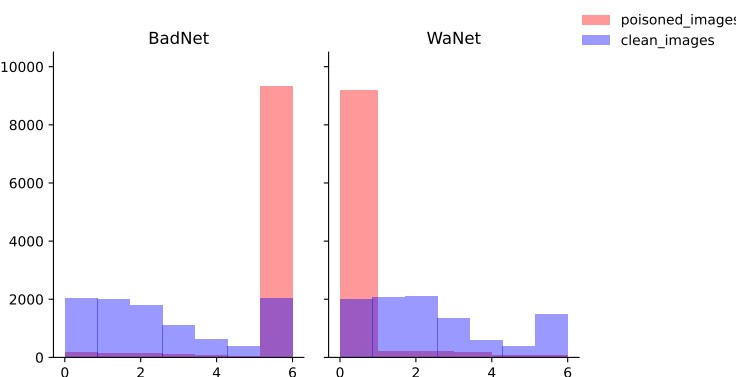

Figure 10: Comparison of FPS score distribution between backdoored and clean images

generating counterfactual samples. To assess the impact of counterfactual sample design, we compare the performance of our method with mixup and random masking on CIFAR-10 dataset and report the results in Table 2. As the table suggests, random noise shows the most stable performance across all types of backdoor attacks. A possible explanation is that random noise enables counterfactual sample generation with finer granularity.

## L   Score distribution

To visually demonstrate the effectiveness of BBCaL, we plot the distribution of the FPS values for backdoor samples and clean samples in Figure 10. As the figure suggests, the FPS values for clean samples center in the middle but those for backdoor samples lie on the two sides, aligning with the causal analysis derived in the last section.

## M   More Details about Defending Multi-Trigger Backdoor Attacks (MTBAs)

We first formulate the MTBAs problem following (Li et al., 2024): For the MTBAs, we randomly selected a subset of training samples, $D_c$, which was then uniformly divided into $m$ smaller subsets, denoted as $D_c = \{D_c^k\}_{k=1}^m$. Each $D_c^k$ received a randomly chosen trigger from a trigger function pool $T = \{\delta_k\}_{k=1}^m$, forming $D_b^k = \{(x_b^k, y_b^k) | x_b^k = x_c^k + \delta_k(x_c^k), x_c^k \in D_c^k\}$. Different target labels were randomly assigned to different triggers. The training model on this multi-trigger poisoned dataset $\{D_b^k\}_{k=1}^m$ is formulated as follows:

$$\min_\theta \mathbb{E}_{(x_i, y_i) \sim D_{/c}}[L(f_\theta(x_i), y_i)] + \sum_{k=1}^m \mathbb{E}_{(x_j, y_j) \sim D_b^k}[L(f_\theta(x_j), y_j)].$$

To evaluate the effectiveness of our detection method in the MTBAs setting, we consider a challenging experiment setup: the trigger pool contains both sample-agnostic and sample-specific triggers. This challenging setting helps us to explore the effectiveness of our model under the mixture of both sample-agnostic and sample-specific triggers. Specifically, we used BadNet and WaNet on CIFAR-10 dataset, which belong to the sample-agnostic and sample-specific attacks, respectively. After backdoor learning over the dataset poisoned with BadNet and WaNet, we obtain a model which achieves 92% in CA, 98% in WaNet ASR, and 100% in BadNet ASR. Then, we run the detection methods on a test dataset which consists of 5000 BadNet-poisoned samples, and 5000 WaNet-poisoned samples. The results are shown in Figure 5.

It is observed that our method still outperforms other baselines, preliminary demonstrating that our method works well in the MTBAs setting. For the causal analysis under the MTBAs setting, the causal structure employed for analyzing MTBAs depends on what trigger is used by the attackers in the inference time. For example, if BadNet is utilized, it corresponds with the causal analysis of backdoored images with sample-agnostic triggers, resulting in a very high FPS. Conversely, if WaNet is used, it aligns with the causal

|  | Precision | Recall | AUROC |
|---|---|---|---|
| **STRIP** | 0.69 | 0.57 | 0.62 |
| **Frequency** | 0.40 | 0.23 | 0.54 |
| **LAVA** | 0.52 | 0.21 | 0.63 |
| **ScaleUP** | 0.68 | 0.61 | 0.70 |
| **Ours** | **0.80** | **0.92** | **0.91** |

Table 5: Performance comparison on defending multi-trigger backdoor attack.

analysis of sample-specific backdoor samples, resulting in a very low FPS. Regardless of the trigger employed, backdoored samples can be effectively distinguished from clean samples.

## N   More Details about Other Potential Adaptive Attack

In this section, we explore another potential adaptive attack, specifically composite backdoor attack, on CIFAR10 dataset. Notably, this attack is successfully injected with an attack success rate of 95% and a clean accuracy of 92%. Below are the comparison results for the detection effectiveness:

|  | Precision | Recall | AUROC |
|---|---|---|---|
| **STRIP** | 0.10 | 0.03 | 0.14 |
| **Frequency** | 0.34 | 0.44 | 0.55 |
| **LAVA** | 0.46 | 0.38 | 0.49 |
| **ScaleUP** | 0.55 | 0.61 | 0.62 |
| **Ours** | **0.70** | **0.62** | **0.76** |

Table 6: Performance comparison on defending composite backdoor attack.

As shown, our method outperforms the other baselines in detecting composite backdoor attacks. However, we also observe that the precision/recall values are lower than those of BadNet or Blend. The explanation is that composite backdoor attacks belong to the category of semantic backdoor attacks, which challenges the effectiveness of our detection method.

## O   Comparison with  (Zhang et al., 2023) and  (Liu et al., 2024)

Causal inference is also used in (Zhang et al., 2023), where they view the backdoor attack as a confounder in the data poisoning process. However, Our causality analysis is fundamentally different from that in (Zhang et al., 2023) with the three aspects:

**Analysis Objective**   Their analysis aims to provide a theoretical analysis for training a clean model from a backdoored dataset, while our analysis aims to investigate the distinct prediction behaviors of clean and backdoored images from a causal perspective.

**Analysis Content**   Their analysis uses causal graphs to model the generation process of backdoor data in the **training stage**, while our analysis focuses on DNNs' prediction behaviors in the **inference stage** when input with different types of data. Although similar structures are used (e.g., $I \leftarrow A \rightarrow Y$), the actual meaning of each edge are fundamentally different. For example, in [1], $A \rightarrow Y$ means backdoor attackers will "change the labels to the targeted label" when constructing backdoor samples (evidenced by §3.2 of (Zhang et al., 2023)), but in our setting, $A \rightarrow Y$ means that backdoor attacks will make the backdoored DNNs predict the input image as label $Y$.

**Analysis Usage** Their analysis aims to adjust confounder, by disentangling backdoor path and causal path in the model training process. However, our causal analysis works as a guideline for distinguishing backdoor samples and clean samples in the inference stage.

(Liu et al., 2024) also explored causality in the fields of language models, where front-door adjustment in causal inference was used to fine-tune a pre-trained language model as the defense model to correct predictions. In addition to the significant differences between fields (backdoor attacks in computer vision and backdoor attacks in NLP), causal analysis was used for different purposes. Hence, this approach is infeasible for our task due to limited information and efficiency requirements.

# P   Proof

We first introduce an important lemma that is key to our proof.

**Lemma 7** (Infinite width networks as linearized networks (Lee et al., 2019)). *Let $f(x)$ denote a fully-connected neural network with $L$ hidden layers, each with width $n_l$ for $l = 1, \ldots, L$. Let $f_t(x) = h^{L+1}(x) \in \mathbb{R}^k$ denote the output of the neural network at time $t$. For a neural network $f_t(x)$ with infinite width, let $\Theta$ denote the corresponding deterministic kernel, and $f_t^{lin}(x)$ denote the linearization of $f_t(x)$. If the learning rate $\eta$ satisfies $\eta < \eta_{critical} := 2 \left( \lambda_{min}(\Theta) + \lambda_{max}(\Theta) \right)^{-1}$, where $\lambda_{min/max}(\Theta)$ are the minimum and maximum eigenvalues of $\Theta$, respectively, then for every $x \in \mathbb{R}^{n_0}$ with $||x||_2 \leq 1$, as $n \to \infty$ and $t \to \infty$, $f_t(x)$ converges in distribution to the same Gaussian distribution as the linearized model $\Theta(\mathcal{X}_T, \mathcal{X})\Theta^{-1}\mathcal{Y}$.*

Our proof is grounded in the intuition that, although directly analyzing a black-box DNN is notoriously challenging, recent studies on the Neural Tangent Kernel (NTK) (Jacot et al., 2018) have shown that a DNN with infinite width can be treated as a linearized network. Following (Guo et al., 2021; 2023), we leverage the RBF kernel and treat the DNN as a $k$-way kernel least square classifier. Under Lemma 7, the regression solution for the NTK is as follows:

$$\phi_q^b(\cdot) = \frac{\sum_{i=1}^{N_b} \Theta(\cdot, \boldsymbol{x}_i) \cdot y_q}{\sum_{i=1}^{N_b} \Theta(\cdot, \boldsymbol{x}_i) + \sum_{j=1}^{N_p} \Theta(\cdot, \boldsymbol{x}_j)}, \tag{9}$$

$$\phi_t^p(\cdot) = \frac{\sum_{i=1}^{N_b} \Theta(\cdot, \boldsymbol{x}_i) \cdot y_t + \sum_{j=1}^{N_p} \Theta(\cdot, \boldsymbol{x}_j) \cdot y_t}{\sum_{i=1}^{N_b} \Theta(\cdot, \boldsymbol{x}_i) + \sum_{j=1}^{N_p} \Theta(\cdot, \boldsymbol{x}_j)}, \tag{10}$$

where $\phi_q^b(\cdot)$ and $\phi_t^p(\cdot) \in \mathbb{R}$ represent the predicted probabilities of class $q$ and class $t$ based on $f(\cdot; \theta)$ for clean samples and backdoored samples, respectively. $N_b$ and $N_p$ denote the number of clean and backdoored samples, respectively. $\boldsymbol{x}_i$ and $\boldsymbol{x}_{iq}$ represent benign samples and benign samples specifically from class $q$, respectively. $\boldsymbol{x}_j$ represents the poisoned samples. $y_q$ and $y_t$ are the corresponding one-hot labels for class $q$ and target class $t$, respectively. The kernel function $\Theta(\boldsymbol{x}, \boldsymbol{x}_i) = e^{-2\gamma||\boldsymbol{x}-\boldsymbol{x}_i||^2}$ ($\gamma > 0$) is used. Given the assumption that training samples are evenly distributed, there are $\frac{N_b}{k}$ benign samples belonging to each class. Without loss of generality, we assume $y_q = 1$ for class $q$ while others are set to 0. The regression solution is then expressed as:

$$\phi_q^b(\cdot) = \frac{\sum_{i=1}^{N_b/k} \Theta(\cdot, \boldsymbol{x}_{iq})}{\sum_{i=1}^{N_b} \Theta(\cdot, \boldsymbol{x}_i) + \sum_{j=1}^{N_p} \Theta(\cdot, \boldsymbol{x}_j)}, \tag{11}$$

$$\phi_t^p(\cdot) = \frac{\sum_{i=1}^{N_b/k} \Theta(\cdot, \boldsymbol{x}_{it}) + \sum_{j=1}^{N_p} \Theta(\cdot, \boldsymbol{x}_j)}{\sum_{i=1}^{N_b} \Theta(\cdot, \boldsymbol{x}_i) + \sum_{j=1}^{N_p} \Theta(\cdot, \boldsymbol{x}_j)}. \tag{12}$$

## P.1   Proof of perturbation on clean images

**Theorem 8** (Prediction consistency for clean samples). *Let $N_b$ denote the total number of benign samples, and let $\frac{N_b}{k}$ represent the number of benign samples in a specific class. Given a well-trained backdoored model*

$f_\theta$ and a clean input $\boldsymbol{x}$ from class $q$, define $\alpha = \mathbb{E}[||\boldsymbol{x} - \boldsymbol{x}_q||^2]]$ and $\beta = \mathbb{E}[||\boldsymbol{x} - \boldsymbol{x}_{/q}||^2]$, where $\boldsymbol{x}_q$ and $\boldsymbol{x}_{/q}$ denote benign samples from class $q$ and benign samples from other class, respectively. Assuming that $\boldsymbol{x}_{/q}$ follows a Gaussian distribution, if we perturb the clean input with scaled Gaussian noise $\boldsymbol{\epsilon} \sim \mathcal{N}(0, a_j I)$, the prediction remains unchanged until $\boldsymbol{\epsilon} > \frac{\alpha^2 + \frac{\ln m}{2\gamma} - \beta^2}{2(\beta - \alpha)}$.

*Proof.* The proof is provided under Lemma 7. After substituting the clean test sample $\boldsymbol{x}$, whose ground truth is class $q$, into Equation 11, we have:

$$\phi_q^b(\boldsymbol{x}) = \frac{\sum_{i=1}^{N_b/k} \Theta(\boldsymbol{x}, \boldsymbol{x}_{iq})}{\sum_{i=1}^{N_b} \Theta(\boldsymbol{x}, \boldsymbol{x}_i) + \sum_{j=1}^{N_p} \Theta(\boldsymbol{x}, \boldsymbol{x}_j)} \tag{13a}$$

$$\approx \frac{\sum_{i=1}^{N_b/k} \Theta(\boldsymbol{x}, \boldsymbol{x}_{iq})}{\sum_{i=1}^{N_b} \Theta(\boldsymbol{x}, \boldsymbol{x}_i)} \tag{13b}$$

$$= \frac{\sum_{i=1}^{N_b/k} e^{-2\gamma||\boldsymbol{x} - \boldsymbol{x}_{iq}||^2}}{\sum_{i=1}^{N_b - N_b/k} e^{-2\gamma||\boldsymbol{x} - \boldsymbol{x}_{i(/q)}||^2} + \sum_{i=1}^{N_b/k} e^{-2\gamma||\boldsymbol{x} - \boldsymbol{x}_{iq}||^2}}, \tag{13c}$$

where the trivial term $\sum_{j=1}^{N_p} \Theta(\boldsymbol{x}, \boldsymbol{x}_j)$ in Equation 13a is omitted following (Guo et al., 2021; 2023), due to the following two reasons: (1) $\boldsymbol{x}$ is a clean sample with low similarity to backdoored samples; (2) Furthermore, for commonly used backdoor attacks, the number of backdoored samples is only a small fraction of that of the clean samples, i.e., $N_p \ll N_b$.

For a well-trained model, the probability of predicting class $q$ for an image from class $q$ should be greater than that for any other class. Therefore, the following equation must hold:

$$\sum_{i=1}^{N_b/k} e^{-2\gamma||\boldsymbol{x} - \boldsymbol{x}_{iq}||^2} - \sum_{i=1}^{N_b - N_b/k} e^{-2\gamma||\boldsymbol{x} - \boldsymbol{x}_{i(/q)}||^2} > 0. \tag{14}$$

For notational convenience, let $\alpha = \mathbb{E}[||\boldsymbol{x} - \boldsymbol{x}_q||^2]$ represent the expected L2-norm of the difference between the input image and the clean images in class $q$, and let $\beta = \mathbb{E}[||\boldsymbol{x} - \boldsymbol{x}_{/q}||^2]$ denote the expected L2-norm of the difference between the input image and the clean images from other classes. Substituting $\alpha$ and $\beta$ into Equation 14 results in the following equation:

$$\frac{N_b}{k} e^{-2\gamma\alpha} - (N_b - \frac{N_b}{k}) e^{-2\gamma\beta} > 0. \tag{15}$$

It is noted that the term $(N_b - \frac{N_b}{k})$ can be simplified to $m\frac{N_b}{k}$. For instance, in a 10-class dataset like CIFAR-10, $m = 9$, and for ImageNet, $m = 99$. Consequently, Equation 15 can be rewritten as:

$$e^{-2\gamma\alpha} - m e^{-2\gamma\beta} > 0. \tag{16}$$

Taking the logarithm of both sides and dividing by $-2\gamma$ gives:

$$\alpha + \frac{\ln m}{2\gamma} - \beta < 0. \tag{17}$$

Now, consider a scenario in which a scaled Gaussian perturbation, $\boldsymbol{\epsilon}$, is added to a clean image. Consequently, Equation 14 becomes:

$$\sum_{i=1}^{N_b/k} e^{-2\gamma||\boldsymbol{x} + \boldsymbol{\epsilon} - \boldsymbol{x}_{iq}||^2} - \sum_{i=1}^{N_b - N_b/k} e^{-2\gamma||\boldsymbol{x} + \boldsymbol{\epsilon} - \boldsymbol{x}_{i(/q)}||^2}. \tag{18}$$

Following a similar procedure as before, we substitute $\alpha' = \mathbb{E}[||\boldsymbol{x} + \boldsymbol{\epsilon} - \boldsymbol{x}_q||^2]$ and $\beta' = \mathbb{E}[||\boldsymbol{x} + \boldsymbol{\epsilon} - \boldsymbol{x}_{/q}||^2]$. We assume that $x_{/q}$ follows a Gaussian distribution. Consequently, we derive the following equation:

$$\alpha' > \alpha; \beta' < \beta. \tag{19}$$

Adding $\epsilon$ to the original image $\boldsymbol{x}$ increases the difference between the new image and its original ground truth, $\boldsymbol{x}_q$. Hence, if $\epsilon$ is larger enough, $\alpha' > \alpha$. Moreover, applying sufficient Gaussian noise can result in an image that essentially becomes pure Gaussian noise Ho et al. (2020). Consequently, the differences between the current image and those from other classes decreases. Following this trend, after adding sufficient noise, $\alpha' + \frac{\ln m}{2\gamma} - \beta' > 0$, whereas the prediction for the original image is $\alpha + \frac{\ln m}{2\gamma} - \beta < 0$. Therefore, the prediction would change. Next, we will identify this threshold. Using $\alpha$ and $\beta$, Equation 18 can be rewritten as:

$$e^{-2\gamma(\alpha+\epsilon^2+2(\boldsymbol{x}-\boldsymbol{x}_q)^T\epsilon)} - m e^{-2\gamma(\beta+\epsilon^2+2(\boldsymbol{x}-\boldsymbol{x}_{/q})^T\epsilon)}. \tag{20}$$

Taking the logarithm of both sides and dividing by $-2\gamma$ yields the following.

$$(\alpha + \epsilon^2 + 2(\boldsymbol{x} - \boldsymbol{x}_q)^T\epsilon)) + \frac{\ln m}{2\gamma} - (\beta + \epsilon^2 + 2(\boldsymbol{x} - \boldsymbol{x}_{/q})^T\epsilon)$$
$$= \alpha + \frac{\ln m}{2\gamma} - \beta + 2(\boldsymbol{x}_{/q} - \boldsymbol{x}_q)^T\epsilon. \tag{21}$$

Since in Equation 17, we have $\alpha^2 + \frac{\ln m}{2\gamma} - \beta^2 < 0$, when $\epsilon$ is small enough ($\epsilon \to 0$), Equation 21 remains less than 0 and Equation 18 is still greater than 0, indicating that the prediction remains unchanged. However, if $\epsilon > \frac{\alpha^2 + \frac{\ln m}{2\gamma} - \beta^2}{2(\beta-\alpha)}$, Equation 21 becomes greater than 0 and Equation 18 becomes less than 0, leading to a change in prediction. This proof sheds light on our causal analysis, demonstrating that clean images experience gradual outcome changes in response to noise intensity.

$\square$

## P.2 Proof of perturbation on Backdoored images

Now we analyze the behavior of backdoor samples after perturbation. For a given backdoored sample, we can simplify Equation 12 as:

$$\phi_t(\boldsymbol{x}) \approx \frac{\sum_{j=1}^{N_p} \Theta(\boldsymbol{x}, \boldsymbol{x}_j)}{\sum_{i=1}^{N_b} \Theta(\boldsymbol{x}, \boldsymbol{x_i}) + \sum_{j=1}^{N_p} \Theta(\boldsymbol{x}, \boldsymbol{x_j})}, \tag{22}$$

Following (Guo et al., 2021; 2023), we omit the term $\sum_{i=1}^{N_b/k} \Theta(\boldsymbol{x}, \boldsymbol{x}_{it})$. This is because $\boldsymbol{x}$ usually does not belong to the target class $y_t$. Hence $\sum_{i=1}^{N_b/k} \Theta(\boldsymbol{x}, \boldsymbol{x}_{it}) \ll \sum_{j=1}^{N_p} \Theta(\boldsymbol{x}, \boldsymbol{x}_j)$. Otherwise, the attacker would not have the motivation to craft backdoored samples.

### P.2.1 Model Agnostic Attack

According to the definition of model agnostic attack in §2, we can express the backdoored sample as $\boldsymbol{x} = \boldsymbol{x'} + \boldsymbol{t}$, where $\boldsymbol{t}$ is a constant value denoting the trigger pattern. Then, similar to the proof process for the clean images, we assume that the backdoored model is well-trained. Hence, for a backdoored image with target label $y_t$, the output probability for the class $y_t$ should be greater than other classes. Therefore, we have:

$$\sum_{j=1}^{N_p} \Theta(\boldsymbol{x}, \boldsymbol{x}_j) - \sum_{i=1}^{N_b} \Theta(\boldsymbol{x}, \boldsymbol{x}_i) > 0$$
$$= \sum_{j=1}^{N_p} e^{-2\gamma||\boldsymbol{x'}+\boldsymbol{t}-(\boldsymbol{x'}_j+\boldsymbol{t})||^2} - \sum_{i=1}^{N_b} e^{-2\gamma||\boldsymbol{x'}+\boldsymbol{t}-(\boldsymbol{x}_i)||^2} > 0. \tag{23}$$

Following a similar proof process as used for clean samples, we define the expectation of the L2-norm of the difference between the input image and the original clean images of poisoned images as $\alpha = \mathbb{E}[||\boldsymbol{x'} - \boldsymbol{x'}_p||^2]$ and the expectation of the L2-norm of the difference between the input image and the benign images as

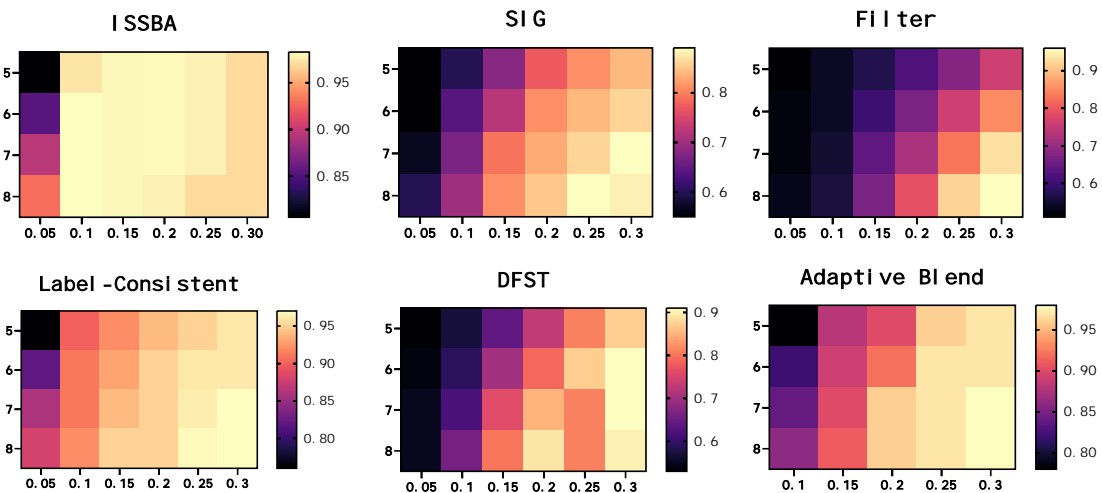

Figure 11: Performance of BBCaL under different combinations of *length* and *step*.

$\beta = \mathbb{E}[||\boldsymbol{x'} + \boldsymbol{t} - \boldsymbol{x}_b||^2]$. This results in $\alpha + \frac{\ln m}{2\gamma} - \beta < 0$. As in the previous proof, after adding the perturbation, the expression becomes $\alpha^2 + \frac{\ln m}{2\gamma} - \beta^2 + 2(\boldsymbol{x}_b - \boldsymbol{x'}_p - \boldsymbol{t})^T \boldsymbol{\epsilon}$. Typically,

$$\mathbb{E}[\boldsymbol{x}_b - \boldsymbol{x'}_p - \boldsymbol{t}] = \mathbb{E}[\boldsymbol{x}_b - \boldsymbol{x'}_p] - \mathbb{E}[\boldsymbol{t}], \tag{24}$$

where $\boldsymbol{x}_b$ and $\boldsymbol{x'}_p$ represent any benign image and the original benign image of a backdoored image, respectively. Since the backdoored image is randomly selected from the benign images, and a trigger is subsequently added, $\boldsymbol{x}_b$ and $\boldsymbol{x'}_p$ are from the same distribution. We assume $\mathbb{E}[\boldsymbol{x_b} - \boldsymbol{x'_p}] = 0$, and since the trigger always adds a positive value, $\boldsymbol{x}_b - \boldsymbol{x'}_p - \boldsymbol{t}$ is always negative. Therefore, $\alpha^2 + \frac{\ln m}{2\gamma} - \beta^2 + 2(\boldsymbol{x}_b - \boldsymbol{x'}_p - \boldsymbol{t})^T \boldsymbol{\epsilon}$ is always less than 0, which implies that after adding perturbation, the prediction remains unchanged.

### P.2.2  Model Specific Attack

According to the definition in §2. For a given backdoored sample with model specific trigger $\boldsymbol{x} = \boldsymbol{x'} + \sigma(\boldsymbol{x'})$, we can simplify eq. (12) as:

$$\sum_{i=1}^{N_p} e^{-2\gamma||\boldsymbol{x'} + \sigma(\boldsymbol{x'}) - (\boldsymbol{x'}_j + \sigma(\boldsymbol{x'}_j))||^2} - \sum_{i=1}^{N_b} e^{-2\gamma||\boldsymbol{x'} + \sigma(\boldsymbol{x'}) - \boldsymbol{x}_i||^2} > 0. \tag{25}$$

Due to the specific nature of the sample-specific attack, where the trigger is tailored to individual images, the trigger for $\boldsymbol{x'}$ becomes ineffective for other new images (e.g., $\boldsymbol{x'} + \boldsymbol{n}$). Hence

$$\sum_{i=1}^{N_p} e^{-2\gamma||\boldsymbol{x'} + \boldsymbol{n} + \sigma(\boldsymbol{x'}) - (\boldsymbol{x'}_j + \sigma(\boldsymbol{x'}_j))||^2} - \sum_{i=1}^{N_b} e^{-2\gamma||\boldsymbol{x'} + \boldsymbol{n} + \sigma(\boldsymbol{x'}) - \boldsymbol{x}_i||^2} < 0, \tag{26}$$

where $\boldsymbol{n}$ is a small change for the input. Hence when $\boldsymbol{\epsilon} > \boldsymbol{n}$, we have that the prediction of the perturbed image would change immediately.

## Q   More Experiments about Magnitude Set

Figure 11 presents additional heatmaps, showing the performance of BBCaL under different combinations of *length* and *step* against different backdoor attacks. In particular, *length* represents the length of the magnitude set $\mathbb{S}$, and *step* represents the difference between two adjacent noise values in the magnitude set.

# R    Impact Statement

Deep neural networks have been widely adopted in a variety of domains, making it crucial to assess their security in practical applications. In this paper, we propose a simple yet effective method for solving the inference-stage black-box backdoor detection under a strict but practical scenario of Model-as-a-Service (MaaS). As outlined in the threat model, our method is proposed from the perspective of a defender. Therefore, this paper has no ethical issues and will not introduce any additional security risks to the DNNs.

