# OpenReview forum: "BBCaL: Black-box Backdoor Detection under the Causality Lens"
_TMLR — Accepted by TMLR_

### Review · Reviewer_PhMm · 2024-10-02

**Summary Of Contributions:**

This paper introduces a novel and intriguing black-box backdoor detection technique from a causality perspective. Specifically, the authors analyze backdoor behavior using causal graphs and observe the different behaviors of backdoor and clean samples under counterfactual interventions. They find that sample-agnostic backdoor attacks tend to maintain robust predictions, whereas sample-specific backdoors become fragile when subjected to varying levels of Gaussian noise as an intervention. Based on these observations, the paper presents a backdoor detection method that compares the prediction robustness of backdoor and clean samples across a series of noise interventions. Clean samples are expected to show a gradual decrease in prediction accuracy, while backdoor samples are anticipated to exhibit either a slight (for sample-agnostic) or significant (for sample-specific) decrease.
The authors conduct extensive experiments across various datasets, models, and backdoor attacks to demonstrate the effectiveness of the proposed method compared to existing baselines. Additionally, the paper provides insightful theoretical proofs supporting the proposed approach.

**Audience:**

Yes

**Claims And Evidence:**

Yes

**Requested Changes:**

The paper is well-written, and the proposed method is both novel and interesting. I appreciate the analysis from a causality perspective to model backdoor behavior. However, there are a few concerns regarding some of the claims and experiments:

(1) Insufficiently Supported Claims:

**Trigger Injection Function:** In Section 3.1, the authors model the trigger injection function as $\hat{x}_i = x_i + t_i$. The "+" operation may not be precise for many backdoor attacks. For instance, in attacks like BadNet and Blend, the trigger often replaces part of the original image rather than merely adding to it. Additionally, triggers in attacks such as WaNet and DFST involve complex transformations.

**Impact on Noise Intervention Assumption:** This imprecision affects the observation in Equation (2), where the authors assume that noise intervention on backdoor samples is equivalent to applying noise to clean samples followed by trigger injection. This assumption may not be rigorous and may not hold in practice.

**Counter-Example with Blend Trigger:** A typical counter-example is the Blend trigger. The authors classify Blend as a sample-agnostic backdoor that is robust against noise intervention. However, if a small, fixed Gaussian noise is used as the trigger pattern (as discussed in the original Blend paper beyond the HelloKitty trigger) and noise intervention is applied afterward, it is likely that the success rates of backdoor samples would decrease. This is because the noise intervention distorts the fixed noise trigger patterns.

**Recommendation**: I suggest the authors elaborate on this point, possibly refining the trigger injection model to better align with various backdoor attack strategies and reassessing the noise intervention assumptions accordingly.

(2) Limited Analysis of Less Effective Cases:

**Filter Attacks:** According to Table 1, Filter attacks are challenging to detect using the proposed method, achieving an AUC of nearly 0.8, although this still outperforms the baselines. The paper does not provide an in-depth analysis of why the proposed method is less effective for Filter attacks
A possible reason is that Filter triggers are typically semantic (e.g., a style transfer) and behave more similarly to natural images compared to other typical backdoor triggers. This similarity makes it difficult to differentiate them from clean samples.

**Recommendation:** I suggest the authors provide insights into why the proposed method struggles with Filter attacks. Understanding the underlying reasons can help in refining the detection technique or guiding future research to address these challenges.

(3) Overlooked Potential Adaptive Attacks:

**Discussion on Adaptive Attacks:** While the authors discuss potential adaptive attacks against the proposed detection method in the last paragraph of Section 5.4, this discussion could be expanded to include more straightforward alternatives.

**Example with Natural Object Triggers:** For instance, a backdoor could leverage natural objects as triggers [1]. In this scenario, the backdoor behavior would involve natural objects rather than out-of-context triggers. Natural object triggers, while serving as triggers, behave more like clean samples and may exhibit similar patterns under noise intervention, potentially evading detection.

**Recommendation:** I suggest the authors provide a discussion on such scenarios where natural object triggers are used. Exploring how the proposed method fares against these adaptive attacks can offer a more comprehensive evaluation of its robustness.

Reference:

[1] Lin, Junyu, et al. "Composite backdoor attack for deep neural network by mixing existing benign features." Proceedings of the 2020 ACM SIGSAC Conference on Computer and Communications Security. 2020.

**Strengths And Weaknesses:**

Strenghts:
1. Novel Perspective: Introduces an innovative approach to backdoor detection using causality, accompanied by clear and intuitive explanations.
2. Effective Technique: Presents an effective and efficient backdoor detection method that outperforms existing baselines.
3. Comprehensive Experiments: Conducts extensive experiments on a variety of datasets, models, and backdoor attack types, ensuring the method's robustness and generalizability.
4. Theoretical Support: Provides insightful theoretical proofs that support the proposed detection technique.

Weaknesses:
1. Insufficiently Supported Claims: Some claims lack robust support and may not fully align with empirical results.
2. Limited Analysis of Less Effective Cases: The paper does not delve deeply into cases where the proposed method is less effective.
3. Overlooked Potential Adaptive Attacks: Potential adaptive attacks against the detection method are not thoroughly explored.

---

> ### Author Response · Authors · 2024-10-16
> **Response to Reviewer PhMm (Part 1/2)**
>
> We are grateful for your detailed requested changes, which have significantly improved the paper. We have made ALL the requested changes in our latest version and highlighted them in blue.
>
> **1. Insufficiently Supported Claims: Some claims lack robust support and may not fully align with empirical results.**
>
> Thanks for the insightful question. We initially use $\delta(x_i) = t_i$ as backdoor triggers, which is indeed a simplified modeling for backdoor attacks. Therefore, we modify the modeling part and adopt a more general setting following the previously well-established works[1-3]:
>
> $\hat{x}_i = x_i + \delta(x_i)$
>
> Here $\delta(x_i)$ can be seen as a trigger generation function based on the given image $x_i$, which varies under different backdoor attacks. For example in BadNets attack [4], $\delta(x_i) = (t - x_i) \odot m$, where $t$ is the square-style trigger pattern and $m$ is a binary mask denoting the location of the trigger pattern. In Blend attack [5], $\delta(x_i) = t$, where $t$ denotes the global trigger pattern such as the random noise.
>
> In this way, we modify the following parts correspondingly to align with the updated backdoor attack modeling:
>
> - Definition 1:
> The trigger function $\delta({x}_i)$ of sample-agnostic backdoor attacks contain the same $t$, for all $x_i$.
>
> - Definition 2:
> The trigger function $\delta({x}_i)$ of sample-specific backdoor attacks contain different $t$ for different $x_i$.
>
> - Equation (2):
> $
> \tilde{x}_i^j = \hat{{x}}_i + {n}_j= {x}_i + \delta(x_i) + {n}_j  = ({x}_i + {n}_j) + \delta(x_i) = {x}''_i + \delta(x_i).
> $
>
>
> These changes would largely help to improve our writing part. We'd also like to emphasize that, these changes would not influence the subsequent causal analysis, intuitions, and methodologies presented in this paper. **Thank you for your insightful suggestion, we have highlighted the corresponding modifications in blue.**
>
>
> **2. Limited Analysis of Less Effective Cases: Filter Attacks**
>
> Thanks for the insightful question, **we have added the discussion on filter attack in the main text as well as the Appendix I highlighted in blue**. Our explanations on why our method is less effective for Filter backdoor attacks are similar to your initial thought. For semantic backdoor attacks such as Filter, DFST, and Composite, they all rely on semantic features as a trigger to launch backdoor attacks. For example, Filter relies on a specific filter, DFST relies on a style transfer learned from a couple of sunset images, and Composite relies on compositional semantic features from other classes. Therefore, when running our detection method over these backdoor attacks, it is observed that the distributions of the backdoor samples' FPS scores tend to have a closer distribution to the clean samples' FPS scores. Specifically, (1) when we start adding random noise, the prediction label of these backdoor samples will not immediately change; (2) when we gradually add larger magnitudes of random noise, the prediction label of these backdoor samples will ultimately change with a faster speed than the clean samples, potentially because the semantic features in the trigger are not as complex as those in the clean image. Thereby, observation (1) makes the FPS not as low as the typical sample-specific backdoor attacks such as WaNet; and observation (2) makes the FPS not as stably high as the typical sample-agnostic backdoor attacks such as BadNets. Therefore, this makes our method seemingly less effective for semantic backdoor attacks.
>
> However, we'd like to emphasize that **(1)** although it is slightly less effective, our method still outperforms the baselines, and **(2)** this discussion poses a really noteworthy question that might be helpful for further refining our method. Inspired by this interesting question, we'd like to leave more explorations in the future works.

---

> ### Author Response · Authors · 2024-10-16
> **Response to Reviewer PhMm (Part 2/2)**
>
> **3. Overlooked Potential Adaptive Attacks: Potential adaptive attacks against the detection method are not thoroughly explored.**
>
> Thank you for your insightful question. In accordance with your suggestions, **we have conducted experiments on composite backdoor attacks using CIFAR-10 dataset and have included a detailed discussion in the main text.** This attack was successfully injected with an attack success rate of 95% and a clean accuracy of 92%. The comparison results for detection effectiveness are presented below:
>
> |               | Precision | Recall | AUROC|
> | :---------------- | :------: | ----: | ----: |
> | STRIP       |    0.10  | 0.03  | 0.14 |
> | Frequency           |  0.34  | 0.44 | 0.55 |
> | LAVA    |   0.46  | 0.38 | 0.49
> | ScaleUP |   0.55   | 0.61  |  0.62
> | **Ours** | **0.70** | **0.62** | **0.76**
>
> As shown, our method outperforms the other baselines in detecting composite backdoor attacks. However, we also observe that the precision/recall values are lower than those of BadNet or Blend.  The explanation is similar to Q2: composite backdoor attacks belong to the category of semantic backdoor attacks, which challenges the effectiveness of our detection method.
>
> [1] Guo J, Li Y, Chen X, Guo H, Sun L, Liu C. Scale-up: An efficient black-box input-level backdoor detection via analyzing scaled prediction consistency. arXiv preprint arXiv:2302.03251. 2023 Feb 7.
>
> [2] Guo J, Li A, Liu C. Aeva: Black-box backdoor detection using adversarial extreme value analysis. arXiv preprint arXiv:2110.14880. 2021 Oct 28.
>
> [3] Doan K, Lao Y, Zhao W, Li P. Lira: Learnable, imperceptible and robust backdoor attacks. InProceedings of the IEEE/CVF international conference on computer vision 2021 (pp. 11966-11976).
>
> [4] Gu T, Dolan-Gavitt B, Garg S. Badnets: Identifying vulnerabilities in the machine learning model supply chain. arXiv preprint arXiv:1708.06733. 2017 Aug 22.
>
> [5] Chen X, Liu C, Li B, Lu K, Song D. Targeted backdoor attacks on deep learning systems using data poisoning. arXiv preprint arXiv:1712.05526. 2017 Dec 15.

---

> > ### Comment · Reviewer_PhMm · 2024-10-21
> > **Thanks for the rebuttal**
> >
> > I appreciate the authors' effort during the rebuttal. My concerns have been largely addressed, and the revised version appears more rigorous.
> >
> > While the proposed method may be less effective against certain advanced attacks, it does have merits and provides valuable insights.

---

> > > ### Author Response · Authors · 2024-10-21
> > > **Thanks for the Review**
> > >
> > > Thanks again for reviewing the paper! Your suggestions and encouragement are really valuable to us!

---

### Review · Reviewer_7cR2 · 2024-10-04

**Summary Of Contributions:**

Summary:
This paper studies the black-box backdoor detection problem by conducting do-calculus. Specifically, the authors focus on inference-stage black-box backdoor detection as a firewall to filter out the backdoor inputs in the inference stage. By leveraging the do-calculus in the causal structural model, the clean examples and backdoored examples act differently when facing various magnitudes of perturbation. Under theoretical and empirical analyses, it is shown that noise-augmented sample generated from clean data shows more consistent predictions compared to the backdoored data, therefore, successfully detecting the backdoor examples. Extensive experiments show the effectiveness of the proposed method.

**Audience:**

Yes

**Broader Impact Concerns:**

No ethical concerns.

**Claims And Evidence:**

Yes

**Requested Changes:**

Please see weaknesses part.

**Strengths And Weaknesses:**

Strengths:
- This paper is well-written and easy-to-follow. The motivation is clear, and the proposed method is intuitive and straightforward.
- The proposed method is efficient to implement, and the effectiveness is great, as shown by the quantitative results compared to other baseline methods.

Weaknesses:
- Although this paper considers both sample-agnostic and sample-specific types for backdoor triggers, diverse or adaptive trigger patterns are not explored, which could largely affect the efficacy of the proposed method. It is suggested to discover how the causal struct is for the adaptive triggers and discuss how can the proposed method solve such problems.


- The feasibility of the proposed causal graph is doubtful: Even though the proposed method works well under the given graph assumption, whether it is feasible for real-world scenarios is still unclear. It would be nice to provide more intuitive examples and explanations.

- While the paper claims to be the first to provide a causality analysis for backdoor attacks in the inference stage, many works that conduct causal analysis for attack, defense, and detection are already proposed, it would be nice to incorporate them in the related work.

	- Huang et al., Harnessing out-of-distribution examples via augmenting content and style, in ICLR 2023.
	- Tople et al., Alleviating Privacy Attacks via Causal Learning, In ICML 2020.
	- Ren et al., Towards Interpretable Defense Against Adversarial Attacks via Causal Inference, In MIR 2022.

---

> ### Author Response · Authors · 2024-10-16
> **Response to Reviewer 7cR2 (Part 1/2)**
>
> We are grateful for your detailed requested changes, which have significantly improved the paper. We have made ALL the requested changes in our latest version and highlighted them in blue.
>
> **1. More Explorations and Explanations on the Diverse or Adaptive Trigger Patterns.**
>
> Thank you for highlighting the interesting trigger concept. Upon reviewing the relevant literature, we have not found papers proposing this specific attack. We believe you might be referring to Multi-Trigger Backdoor Attacks (MTBAs), where multiple adversaries employ diverse types of triggers (e.g., both sample-specific and sample-agnostic) to poison the same dataset, as outlined in reference [1]. We also include these experiments in Appendix M, highlighted in blue.
>
> **Problem Setup**: According to these settings, we first formulate the MTBAs problem as follows:
>
> > For the MTBAs, we randomly selected a subset of training samples, $D_c$, which was then uniformly divided into $m$ smaller subsets, denoted as $D_c = \\{  D_c^{k}  \\}\_{k=1}^{m}$. Each $D_c^{k}$ received a randomly chosen trigger from a trigger function pool $T = \\{ \delta_{k} \\}\_{k=1}^{m}$, forming $D_{b}^{k} = \\{(x_b^k, y_b^k) | x_b^k = x_c^k + \delta_k(x_c^k), x_c^k \in D_{c}^{k}\\}$. Different target labels were randomly assigned to different triggers. The training model on this multi-trigger poisoned dataset $\\{D_{b}^{k}\\}\_{k=1}^{m}$ is formulated as follows:
>
> > $\min_{\theta} \mathbb{E}_{(x_i, y_i) \sim D\_{/c}}[L(f\_\theta(x_i), y\_i)] + \sum\_{k=1}^{m} \mathbb{E}\_{(x\_j, y\_j) \sim D\_b^{k}} [L(f\_\theta(x\_j), y\_j)].$
>
> **Experiment**: To evaluate the effectiveness of our detection method in the MTBAs setting, we consider a challenging experiment setup: the trigger pool contains both sample-agnostic and sample-specific triggers. This challenging setting helps us to explore the effectiveness of our model under the mixture of both sample-agnostic and sample-specific triggers. Specifically, we used BadNet and WaNet on CIFAR-10 dataset, which belong to the sample-agnostic and sample-specific attacks, respectively. After backdoor learning over the dataset poisoned with BadNet and WaNet, we obtain a model that achieves 92\% in CA, 98\% in WaNet ASR, and 100\% in BadNet ASR. Then, we run the detection methods on a clean test dataset and a poisoned test dataset which consists of 5000 BadNet-poisoned samples, and 5000 WaNet-poisoned samples. The results are shown below:
>
> |               | Precision | Recall | AUROC|
> | :---------------- | :------: | ----: | ----: |
> | STRIP       |  0.69 |	0.57 |  0.62 |
> | Frequency           |  0.40  | 0.23  | 0.54 |
> | LAVA    |   0.52| 0.21 | 0.63 |
> | ScaleUP |   0.68  | 0.61 | 0.70 |
> | **Ours** | **0.80** | **0.92** | **0.91** |
>
> It is observed that our method still outperforms other baselines, preliminarily demonstrating that our method works well in the MTBAs setting.
>
> **Analysis**: For the causal analysis under the MTBAs setting, the causal structure employed for analyzing MTBAs depends on what trigger is used by the attackers in the inference time. For example, if BadNet is utilized, it corresponds with the causal analysis of backdoored images with sample-agnostic triggers, resulting in a very high FPS. Conversely, if WaNet is used, it aligns with the causal analysis of sample-specific backdoor samples, resulting in a very low FPS. Regardless of the trigger employed, backdoored samples can be effectively distinguished from clean samples. We have included these findings in the experimental part.
>
> If this does not correspond to the trigger you were mentioning, please let us know. We would be delighted to expand our discussion further and address any additional concerns you might have.
>
>
> **2. Feasibility of Causal Graphs for Real-world Scenarios**
>
> Thanks for the inspiring question. We first kindly argue that our causal graphs have not brought additional assumptions on the backdoor attacks, but just rely on the popular widely-recognized concepts of backdoor attacks. In causal inference, a causal graph is typically constructed based on prior knowledge of the relationships between variables, as suggested by domain experts [2-3]. Similarly, drawing upon the widely accepted classification of backdoor attacks [4] and the detailed generation processes for different attacks, we have developed a causal graph specifically tailored for backdoor samples with both sample-agnostic and sample-specific triggers.
>
> Secondly, these causal graphs can cover a wide range of backdoor attacks proposed in the previous research [4]. For example, the nine well-established real-world attacks discussed in our paper have empirically demonstrated adaptability to this causal graph, as evidenced by the advanced performance metrics obtained.

---

> ### Author Response · Authors · 2024-10-16
> **Response to Reviewer 7cR2 (Part 2/2)**
>
> **3. More Discussion with Related Works**
>
> Thank you for providing us with those interesting and relevant works. We believe that including them has strengthened our paper, and we have incorporated all of them in the updated manuscript.
>
>
> [1] Li Y, Ma X, He J, et al. Multi-Trigger Backdoor Attacks: More Triggers, More Threats[J]. arXiv preprint arXiv:2401.15295, 2024.
>
> [2]Guo R, Cheng L, Li J, et al. A survey of learning causality with data: Problems and methods[J]. ACM Computing Surveys (CSUR), 2020, 53(4): 1-37.
>
> [3] Yao L, Chu Z, Li S, et al. A survey on causal inference[J]. ACM Transactions on Knowledge Discovery from Data (TKDD), 2021, 15(5): 1-46.
>
> [4]Li Y, Jiang Y, Li Z, Xia ST. Backdoor learning: A survey. IEEE Transactions on Neural Networks and Learning Systems. 2022 Jun 22;35(1):5-22.

---

> > ### Comment · Reviewer_7cR2 · 2024-11-04
> > **Thanks for your detailed rebuttal**
> >
> > Dear authors,
> > I have read your detailed rebuttal, which addressed most of my concerns. I am happy to see that diverse types of triggers are incorporated, the effectiveness of the proposed method is again justified, and realistic concerns are also addressed. The revised version incorporated more related references and is shown to be more rigorous.
> > For now, no further concerns from me. Thanks again for your effort.
> >
> > Best.

---

### Review · Reviewer_Wr5G · 2024-10-25

**Summary Of Contributions:**

This work aims to detect backdoor attacks by leveraging causality and spurious correlations. The authors clearly define the problem and propose a noise injection method on images to effectively tackle black-box backdoor detection. The method's effectiveness is validated across multiple datasets and attack types.

**Audience:**

Yes

**Claims And Evidence:**

Yes

**Requested Changes:**

See weakness.

**Strengths And Weaknesses:**

Strengths:
- The paper is well-written, providing a clear formulation of the problem setting, research goal, and review of existing works. Additionally, the formulation of backdoor attacks using causal inference is both insightful and innovative.
- The method is well-motivated, with a logical approach to addressing the research question.
- The authors consider both basic attacks (e.g., BadNets) and more advanced attacks (e.g., WaNet), which demonstrates a comprehensive evaluation.

Weakness:
- Approaching backdoor detection from the perspective of spurious paths is not novel, as similar ideas have been explored in prior works [1, 2, 3]. For example, [3] identifies backdoors by shuffling neural network weights (injecting noise into the weight space). Additionally, [2, 3] discuss the concept of backdoor features as "shortcut features." While the proposed method focuses on a black-box setting, where existing methods may not directly apply, it would be beneficial to acknowledge and discuss these related approaches.

[1] Randomized Channel Shuffling: Minimal-Overhead Backdoor Attack Detection without Clean Datasets.
[2] Indiscriminate poisoning attacks are shortcuts.
[3] Poisons that are learned faster are more effective.

---

> ### Author Response · Authors · 2024-11-03
> **Respond to Reviewer Wr5G**
>
> Dear Reviewer wr5G,
>
> Thank you for recognizing the strengths of our paper and for suggesting these insightful works. We agree they are highly relevant, and we have included them in our latest version to enhance the depth and completeness of our study.
>
> We would also like to take this opportunity to clarify our concept of the “spurious path.” We understand your concern that previous studies have explored framing a backdoor attack as a shortcut. However, while these studies have empirically observed the phenomenon through extensive experimentation, a theoretical understanding of the underlying mechanism has remained limited. Our unique contribution in this paper is to provide a theoretical explanation for the shortcut phenomenon through causal inference. By applying causal inference, we offer a new theoretical lens for understanding this phenomenon, particularly through the causal inference definition of the "spurious path."
>
> Thank you again for the valuable references. We believe their inclusion has further strengthened our manuscript, and we greatly appreciate your thoughtful feedback.
>
> Best regards,
>
> The Authors

---

> > ### Comment · Reviewer_Wr5G · 2024-11-04
> >
> > Thank you, authors, for your response. My concerns have been fully addressed.

---

### Decision · Action_Editor_2DTr · 2024-11-28

**Recommendation:** Accept with minor revision

**Comment:**

This paper studies black-box backdoor detection at inference time. This is the problem where one must detect (and filter out) "backdoor inputs" having access only to input samples and predicted labels. They propose a causal approach to this problem based on the insight that predictions to backdoor samples follow a "spurious path" whereas those for clean samples follow the "causal path". Their black box detection method derived from that insight, aims to distinguish between clean samples and backdoor samples by inspecting the prediction consistency when progressively adding noise. They find empirically that this method can distinguish between clean and backdoor samples for a broad range of attacks. They also provide theoretical analysis.

The reviewers found the paper "well-written" (Reviewer Wr5G, Reviewer 7cR2), "accompanied by clear and intuitive explanations" (Reviewer PhMm); the problem formulation "insightful and innovative" (Reviewer Wr5G); the proposed method "well-motivated" (Reviewer Wr5G), "intuitive and straighforward" (Reviewer 7cR2) and "innovative, effective and efficient " (Reviewer PhMm). The reviewers also found the empirical evaluation "comprehensive", covering basic and more advanced attacks (Reviewer Wr5G), the "effectiveness is great, as shown by the quantitative results compared to other baselines" (Reviewer 7cR2). Reviewer PhMm also recognized the "extensive experiments on a variety of datasets, models, and backdoor attack types, ensuring the method's robustness and generalizability."

Reviewers' concerns regarding missing related work (Reviewer Wr5G, Reviewer 7cR2) have been addressed during the rebuttal; though see the note below in "requested changes".
Other concerns raised relate to other types of trigger patterns that the authors didn't explore (Reviewer 7cR2, Reviewer PhMm), feasibility of causal graph assumptions (Reviewer 7cR2), discussion of limitations and hypotheses for why the proposed approach works less well for some types of attacks, though still outperforming baselines (Reviewer PhMm). The authors have addressed such concerns to the reviewers' satisfaction, by including clarifications of assumptions, running additional experiments, and discussing limitations.

Based on the above, I recommend to accept the paper with a minor requested change, outlined below. I also recommend a featured certification as all reviewers pointed out the innovative perspective of the paper, strong empirical results as well as theoretical analysis. During the rebuttal, the authors addressed various other concerns about more advanced attacks and ran additional experiments.

Requested change:

In Section 2, the authors state: "however, there is limited prior work using causal inference to analyze backdoor attacks during the inference stage. To the best of our knowledge, this paper is the first to provide a causality analysis of backdoor attacks in the inference stage". I think the first of those sentences was added during the rebuttal (after incorporating references suggested by the reviewers) and the second was there from before. Can you please clarify, is this paper indeed the first to do causal analysis of backdoor attacks at the inference stage or is there limited prior work to that? It's okay if this is not the first paper to do so, but in that case please revise claims accordingly for complete accuracy.

**Audience:**

This paper is on the topic of inference-time black box backdoor detection, which is important for enhancing security of machine learning systems. The causal lens adopted here, experiments and theoretical analysis will be of interest to the TMLR community.

**Claims And Evidence:**

This paper studies black-box backdoor detection at inference time. This is the problem where one must detect (and filter out) "backdoor inputs" having access only to input samples and predicted labels. They propose a causal approach to this problem based on the insight that predictions to backdoor samples follow a "spurious path" whereas those for clean samples follow the "causal path". Their black box detection method derived from that insight, aims to distinguish between clean samples and backdoor samples by inspecting the prediction consistency when progressively adding noise. They find empirically that this method can distinguish between clean and backdoor samples for a broad range of attacks. They also provide theoretical analysis.

The authors provide evidence to substantiate their claims, both empirically and theoretically. During the rebuttal, Reviewers 7cR2 and Wr5G challenged the claim that this paper is the first to provide a causality analysis for backdoor attacks in the inference stage. The authors have accordingly revised their related work section to incorporate the works that the reviewer suggested and discussed the differences between their work compared to those papers. The reviewer was satisfied with this discussion. It does sound like, based on the revised related work, the causal model that the authors consider is distinct from prior work. (See below a clarification required in the requested changes). Reviewer PhMm pointed out some other technical claims that need refinement which the authors have also addressed during the rebuttal to the reviewer's satisfaction.

---

> ### Author Response · Authors · 2024-12-18
> **Thanks to AE and Reviewers**
>
> Dear AE and Reviewers,
>
> Thanks so much for your encouragement and suggestions during the paper reviewing period. Our work has improved a lot by incorporating all of your constructive suggestions! Currently, we have submitted a camera-ready version to the system.
>
> **Regarding requested changes**:
> > Q: Can you please clarify, is this paper indeed the first to do causal analysis of backdoor attacks at the inference stage or is there limited prior work to that?
>
> Thank you for your thoughtful question. We would like to clarify that this is, to the best of our knowledge, the first paper to analyze backdoor attacks specifically at the inference stage. The newly included papers [1-3] primarily focus on causality analysis for other well-established problems, such as Out-of-distribution(OOD) detection, adversarial example detection, and privacy attacks. While these are important contributions, they address fundamentally different challenges compared to our work.
>
> Furthermore, within the domain of backdoor attacks and defenses, there are indeed two notable papers that leverage causal analysis, as mentioned in the related works part. However, these studies focus on the training stage, aiming to train a clean model from a backdoored dataset. In contrast, our work investigates the distinct prediction behaviors of clean versus backdoored images at the inference stage.
>
> Based on this distinction, we respectfully maintain that our study is the first to conduct a causal analysis of backdoor attacks at the inference stage.
>
> We acknowledge that the writing was kind of inconsistent in the previous version. We have revised the text accordingly:
>
> 1. "However, there is limited prior work using causal inference to analyze backdoor attacks during the inference stage." -> "However, their causality analysis focuses on other well-established problems, such as OOD detection, adversarial example detection, and privacy attacks, which are fundamentally different from our backdoor detection problem."
> 2. "To the best of our knowledge, this paper is the first to provide a causality analysis of backdoor attacks in the inference stage." -> We slightly changed the sentence order for clarity; more details can be found in the camera-ready version.
>
> Please contact us if any other changes are further required. Thanks again for all of your efforts in the paper reviewing process!
>
> [1] Huang et al., Harnessing out-of-distribution examples via augmenting content and style, in ICLR 2023.
>
> [2] Tople et al., Alleviating Privacy Attacks via Causal Learning, In ICML 2020.
>
> [3] Ren et al., Towards Interpretable Defense Against Adversarial Attacks via Causal Inference, In MIR 2022.